# The Role of Nature-Based Solutions for Improving Environmental Quality, Health and Well-Being

**Hai-Ying Liu [1,\*], Marion Jay [2] and Xianwen Chen [3,4]**

1 Department of Environmental Impacts and Sustainability, NILU—Norwegian Institute for Air Research, P.O. Box 100, 2027 Kjeller, Norway

2 Adelphi Research gGmbH, Alt-Moabit 91, 10559 Berlin, Germany; jay@adelphi.de

3 Oslo Department, NINA—Norwegian Institute for Nature Research, Sognsveien 68, 0855 Oslo, Norway; xianwen.chen@gmail.com

4 Department of Business Administration, Inland School of Business and Social Sciences, Inland Norway University of Applied Sciences, P.O. Box 400, 2418 Elverum, Norway

\* Correspondence: hyl@nilu.no; Tel.: +47-6389-8048

**Abstract:** Nature-based solutions (NbS) have been positioned and implemented in urban areas as solutions for enhancing urban resilience in the face of a wide range of urban challenges. However, there is a lack of recommendations of optimal NbS and appropriate typologies fitting to different contexts and urban design. The analytical frameworks for NbS implementation and impact evaluation, that integrate NbS into local policy frameworks, socio-economic transition pathways, and spatial planning, remain fragmented. In this article, the NbS concept and its related terminologies are first discussed. Second, the types of NbS implemented in Europe are reviewed and their benefits over time are explored, prior to categorizing them and highlighting the key methods, criteria, and indicators to identify and assess the NbS's impacts, co-benefits, and trade-offs. The latter involved a review of the websites of 52 projects and some relevant publications funded by EU Research and Innovation programs and other relevant publications. The results show that there is a shared understanding that the NbS concept encompasses benefits of restoration and rehabilitation of ecosystems, carbon neutrality, improved environmental quality, health and well-being, and evidence for such benefits. This study also shows that most NbS-related projects and activities in Europe use hybrid approaches, with NbS typically developed, tested, or implemented to target specific types of environmental–social–economic challenges. The results of this study indicate that NbS as a holistic concept would be beneficial in the context of climate action and sustainable solutions to enhance ecosystem resilience and adaptive capacity within cities. As such, this article provides a snapshot of the role of NbS in urban sustainability development, a guide to the state-of-the-art, and key messages and recommendations of this rapidly emerging and evolving field.

**Keywords:** biodiversity; blue infrastructure; climate adaptation and mitigation; ecosystem services; green infrastructure; sustainable development goals; urban sustainability

## 1. Introduction

Climate change and urbanization have resulted in a broad range of societal challenges for urban areas [1], such as the loss or degradation of natural areas, soil sealing, drought, and flooding, which pose further challenges to biodiversity, ecosystem functioning, delivery of the ecosystem services (ES) (e.g., clean air, water, and soil), and consequently human health and well-being [2]. Almost 55% of the world's population lives in urban areas today, which will increase to over 68% by 2050 [3,4]. Thus, two of the main future challenges are to design sustainable cities that can adapt to the changing needs of their inhabitants and to the evolution of environmental conditions. Both the United Nations Habitat III "New Urban Agenda" [5] and the Sustainable Development Goal (SDG) 11 "Make cities and human settlements inclusive, safe, resilient, and sustainable" address the needs for

innovative approaches and solutions in urban management to strengthen urban resilience and sustainably neutralize the negative effects of urbanization and climate change on humans and nature [6]. One promising way to achieve these challenges is to adopt nature-based solutions (NbS) in the management and design of urban areas [7].

The term NbS has been advocated by policymakers, governance bodies, scientists, and non-governmental organizations (NGOs). Different definitions of NbS emerged throughout time, including two of international governmental and non-governmental bodies, which greatly shaped the NbS discourse in recent years, i.e., the International Union for Conservation of Nature's (IUCN) and the European Commission's (EC) definitions [8,9]. However, there is no agreed clarification about the differences and commonalities of these two NbS definitions. In addition, the NbS concept is clearly connected to other concepts, such as ecosystem-based adaptation/mitigation (EbA/EbM), green infrastructure (GI), blue infrastructure (BI), blue/green infrastructure/green/blue infrastructure (BGI/GBI), and ecological engineering (EE). Although the extent of similarities and distinctions of these concepts are being discussed in scientific literature, it is not clear whether NbS is distinctly different from these other concepts either [10].

Regardless of this heterogeneity of definitions, NbS are tested through experimentation, designed, and implemented all around the globe, and as such they have been scientifically assessed. Many studies found that NbS support biodiversity conservation [11–13], generate additional environmental–economic–societal benefits [14], and provide a basis for climate change mitigation and adaptation [15]. Several NbS approaches already exist, such as the NbS impact evaluation framework [16], the NbS handbook [17], the NbS core principles [10], and the NbS global standard [18]. Furthermore, information marketing and collaboration are promoted on the NbS repository, see for example the EU repository of NbS–Oppla [19]. However, the existing conceptual and practical knowledge of NbS still remain fragmented and there is a need to develop a unified framework including its methods, criteria, and indicators to measure NbS implementation in different climatic–environmental–socio-economic conditions, to assess its benefits and trade-offs in diverse structures and configurations (e.g., mix of vegetation and trees, species, shape, spatial distribution of public green space, and vegetation coverage), and to exchange experiences, solutions, and good practices, while at the same time enhancing and fostering market opportunities for innovative companies.

As such, the first part of this paper focuses on the concept of NbS (Section 2). The first objective is thus to review the concept of NbS, including a comparison with other related concepts. We proceed with this objective in four steps: (i) describing and analyzing NbS definitions and primary attributes as proposed by the EC and the IUCN (Section 2.1), (ii) describing and categorizing NbS-related concepts (Section 2.2), (iii) analyzing the frequency of the various NbS-related concepts in scientific literature over time (Section 2.3), and (iv) examining the emergence over time in broader discourses and the geographical coverage for the different NbS-related concepts (Section 2.4) (Figure 1).

The second part of this paper focuses on the implementation and assessment of NbS (Section 3). The second objective is thus to investigate the state-of-the-art of the NbS, involving two dimensions: global frameworks of NbS governance (Section 3.1), and implementation and assessment of NbS (Section 3.2). For the governance dimension, we focus on three aspects: description of international intra-governmental and non-governmental frameworks (Section 3.1.1), EU governmental framework (Section 3.1.2), and Chinese governmental framework (Section 3.1.3). For the implementation and assessment dimension, we address four aspects, including (i) a set of criteria used to characterize NbS and the type of NbS implemented (Section 3.2.1), (ii) challenges addressed by the NbS and the key performance indicators (KPIs) to assess NbS impact (Section 3.2.2), (iii) NbS data and metadata platforms (Section 3.2.3), and (iv) NbS stakeholders and their priorities for NbS implementation (Section 3.2.4) (Figure 1).

Section 4 discusses the main issues raised in Sections 2 and 3 and suggests areas where further development is needed. Section 5 offers conclusions.

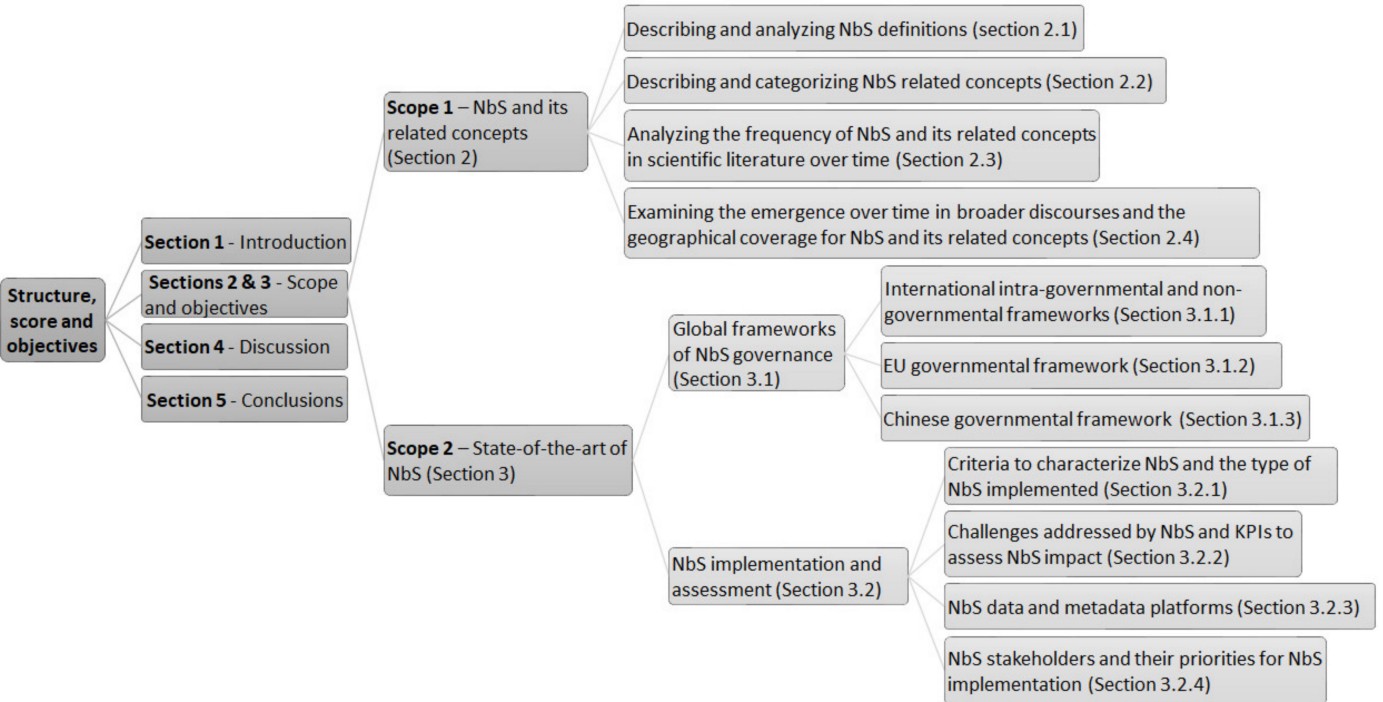

**Figure 1.** Structure, scope, and objectives of the paper.

## 2. An Analysis of NbS Terminologies

### 2.1. Nature-Based Solutions Timeline and Definitions

The term NbS first emerged in the early 2000s, in the discussions on land-use management and planning and water resource management, including the use of wetlands for wastewater treatment [20]. From the mid-2000s, the NbS also began to appear in literature on industrial design [21]. In the year 2008, the NbS were first used by the World Bank to promote nature as a source of solutions to challenges associated with climate change [22,23]. In the year 2009, the IUCN referred to NbS in a position paper for the United Nations (UN) Framework Convention on Climate Change. Afterwards, the term NbS was quickly taken up, supported, and broadened by the IUCN and by the EC [8,10,18,24–28], and became more widely used in literature relating to methods for increasing resilience to the impacts of climate change. The EC has developed an EU Research and Innovation (R&I) agenda on NbS in its Seventh Framework Program (FP7) and H2020 Framework Program [9,29], and is continually addressing NbS in its Green Deal Calls [30] and Horizon Europe Calls [31]. The timeline shown in Figure 2 highlights the major milestones in the evolution of the NbS concept, starting from the first use of the term NbS, to the NbS at the core of the EU R&I agenda.

So far, research on the NbS and its related concept is still very limited, but it has been diversified. In the USA, for example, nature-based infrastructure [32] and engineering with nature [33] are the two common concepts, which overlap with the concept of NbS in several aspects. There are also some differences in the definition of NbS between the EC and the IUCN. The IUCN in the year 2016 [8] (p. 166) defined NbS as "actions to protect, sustainably manage, and restore natural or modified ecosystems, that address societal challenges effectively and adaptively, simultaneously providing human well-being and biodiversity benefits". The EC first defined NbS in the year 2015 [9] (p. 24) as "NbS are inspired and supported by nature and simultaneously provide environmental, social, cultural and economic benefits", and then revised its definition in the year 2019 [34] as "NbS are inspired and supported by nature, which are cost-effective, simultaneously provide environmental, social and economic benefits and help build resilience. Such solutions bring more, and more diverse, nature and natural features and processes into cities, landscapes

and seascapes, through locally adapted, resource-efficient and systemic interventions". For the EC, NbS are understood as "living solutions" aimed at helping society to cope with various environmental, social, and economic challenges in a sustainable manner. They are inspired by nature, supported by nature, or copies from nature [35]. The IUCN defined NbS as "actions" and believes that the basic approach to NbS is to actively apply sustainable management and conservation of natural resources to meet major societal challenges, such as climate change, food security, water security, and natural disasters [7,8].

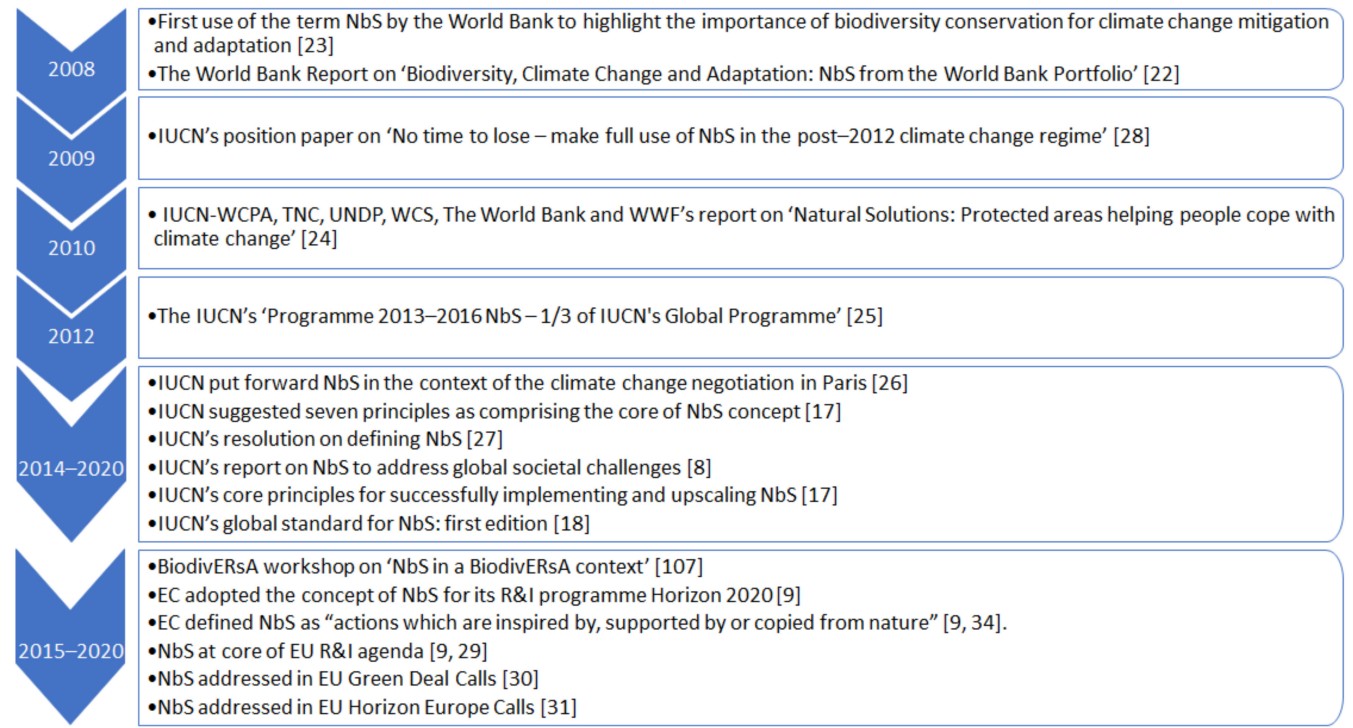

**Figure 2.** Timeline of the development of the nature-based solutions (NbS) concept.

To compare, the IUCN's definition is concerned with human well-being and biodiversity, while the EC's definition explicitly considers cost-effectiveness, resource-use efficiency, and economic benefits, which were not mentioned in the IUCN definition. Although the IUCN's definition does not explicitly state cost-effectiveness, resource-use efficiency, or economic benefits [8,18,36], the goals of "sustainably manage" and "effectively and adaptively" may suggest that the IUCN's definition is concerned with cost-effectiveness, efficiency, and economics, depending on one's definition and interpretation of "sustainably" and "effectively". Considering the following, if an NbS is not cost-effective, efficient, or beneficial, the NbS is likely not going to be implemented and sustainably managed.

In addition, biodiversity and its benefits are not only important for humans, but also for other species such as plants, insects, animals, and microbiota. Biodiversity and its benefits, which is an important goal of utilizing NbS according to the IUCN's definition, is not explicitly included in the EC's definition. Instead, the EC definition includes an explanation of biodiversity as "more, and more diverse, nature, and natural features and processes" which indicates that NbS must therefore benefit biodiversity and support the delivery of a range of ES as well [34,37].

Furthermore, geographically and demographically, the EC's definition for NbS emphasizes on more sustainable and resilient societies through growth and job creation, mostly in urban contexts. It is unclear whether the EC's NbS definition considers rural areas, although rural areas may be included in the broad interpretation of "landscapes and

seascapes". The IUCN's definition was developed from a global perspective, which does not have a specific geographical or demographic focus.

To summarize, we find that the IUCN's NbS definition and the EC's NbS definition are worded differently, but cover similar aspects, both defining NbS as living solutions or actions that utilize nature to deliver multiple benefits and address multiple challenges in a broad way.

### 2.2. Nature-Based Solutions-Related Concepts

The concept of NbS is very closely related to the concepts of EE, GI, BI, BGI/GBI, and EbA/EbM. Table 1 shows the definition and key references of these concepts. We can see that it is not quite clear whether NbS is distinctly different from these other concepts, and a range of crossovers appears throughout the concepts.

The concepts related to the GI, BI, and BGI/GBI are the targeted approaches to specific activities or land use problems. The GI mainly refers to the land, including land reserves, farmland for intensive agriculture land, ecological corridors and underground tunnels built for animals, as well as urban parks and green roofs. The BI is mainly related to water, including coastal areas, rivers, lakes, waterway channels, wetlands, floodplains, and some human design elements (e.g., artificial channels, ponds, reservoirs, and urban sewage). Generally, BI always appears with GI, so it is commonly considered jointly under the headings of BGI or GBI. The green and blue elements of this infrastructure are considered as natural elements that can bring ecological, economic, and social benefits. These concepts also help to understand the benefits that nature brings to human society.

The ecosystem-based concepts EbA and EbM recognize that human and cultural diversity are integral parts of the ecosystem and that conservation and utilization need to be balanced. The EbA and EbM are about ecosystem-based strategies for adapting and mitigating to climate change, respectively. Generally, EbA and EbM appear as one term as EbA/EbM [38]. The protection of ecosystem structure and function and the maintenance of ES should be the priority objectives of the ecosystem approach, and the ecosystem must be managed within its functional scope [39]. The ecosystem-based concepts can guide and realize the fair management of natural resources, to reflect and maintain different needs and values. The EbA/EbM concept also more explicitly involves the objective of addressing climate changes and supporting adaptation and mitigation through natural solutions. In a broader context, the concept and practice of ecological restoration can be linked here as EE. The EE approach, which covers a wide range of activities and interventions, obviously seeks to supplement technology-based infrastructure with natural alternatives and can therefore be considered as an application of NbS.

**Table 1.** Selected terminologies and their definitions found in the literature related to nature-based solutions.

| Terminology | Definition | Key References |
| --- | --- | --- |
| Nature-based Solutions (NbS) | Refers to the sustainable management and use of nature for tackling socio-environmental challenges, including issues such as climate change, biodiversity degradation, water security, water pollution, food security, human health, and disaster risk management. | [8,9,34] |

**Table 1.** *Cont.*

| Terminology | Definition | Key References |
|---|---|---|
| Ecological Engineering (EE) | Sustainable ecosystem designed for the common interests of human society and natural environment is to integrate society with its natural environment. | [40,41] |
| Green Infrastructure (GI) | A strategically planned network of natural and semi-natural areas with other environmental features designed and managed to deliver a wide range of ES. | [42,43] |
| Blue Infrastructure (BI) | Refers to urban infrastructure relating to water bodies, defined as a network providing the "ingredients" for solving urban and climatic challenges by building with nature. | [44] |
| Blue/Green infrastructure/Green/Blue Infrastructure (BGI/GBI) | An interconnected network of natural and designed landscape components, including water bodies and green and open spaces, which provide multiple functions. | [45,46] |
| Ecosystem-based Adaptation/Mitigation (EbA/EbM) | The use of biodiversity and ecosystem services as part of an overall adaptation/mitigation strategy to help people to adapt/mitigate the adverse effects of climate change. Policies and measures that consider the role of ES in reducing the vulnerability of society to climate change, in a multi-sectoral and multi-scale approach. | [47–50] |

## 2.3. Temporal Analysis of the Literature for Nature-Based Solutions and Their Related Concepts

The temporal analysis aimed to analyze the frequency of the NbS and their related concepts in scientific literature over time. It was obtained by using search string in Google Scholar between 2008 and 2020, using the following key words in title-abstract-keywords respectively, i.e., 'nature-based solutions', 'ecological engineering', 'green infrastructure', 'blue infrastructure', 'blue/green infrastructure' or 'green/blue infrastructure', 'ecosystem-based adaptation' or 'ecosystem-based mitigation'. Each provided 10,860, 130,560, 58,730, 2999, 2004, and 5676 hits, respectively. At the end, a total of 210,829 scientific papers, published between 2008 and 2020, which contained any of the concepts listed in Table 1 in their title, keywords, or abstract were obtained from Google Scholar on 18 May 2021 (the day we implemented the temporal analysis). The objective of this temporal analysis is to analyze the frequency of occurrence for NbS and their related concepts found in the literature, and therefore it was not necessary to review the papers in detail. Figure 3 summarizes the frequency of their use, which illustrates time-series trends as well. As expected, as a relatively new ecological term, NbS is steadily rising, with a recent and intense rise since 2016. NbS has been more frequently used in contrast to EbA/EbM,

BI, and BGI/GBI since 2016. GI and EE are much more frequently used in contrast to other terms.

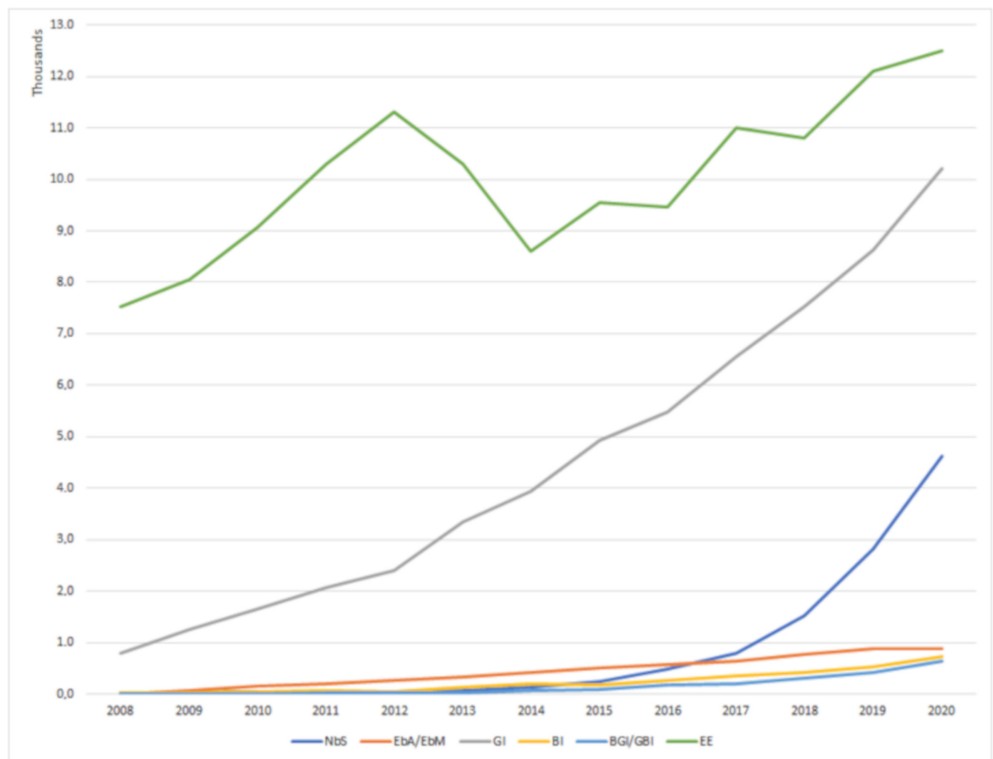

**Figure 3.** The frequency of occurrence of different terms found in the literature relating to nature-based solutions.

### 2.4. Google Trends Analysis for Nature-Based Solutions and Their Related Concepts

Google trends analysis was carried out to examine the emergence of NbS and their related concepts over time in broader discourses and their geographical coverage. Google Trends [51] allows for the examination of relative search volumes of terms over time to illustrate trends in popularity of terms that are more mainstream than academic and is an indicator of movements from the academic literature to more layman outlets, e.g., through media and into popular science.

Google trends analysis was performed by using search strings in Google Trends between 2004 (which is around the time when Google Trends started) and 18 May 2021 (the day we implemented the Google Trends analysis), using the following keywords (listed in Table 1) respectively, i.e., 'nature-based solutions', 'ecological engineering', 'green infrastructure', 'blue infrastructure', 'blue/green infrastructure' or 'green/blue infrastructure', 'ecosystem-based adaptation' or 'ecosystem-based mitigation'. Figure 4 shows the trend for the terms 'GI', 'BI', 'NbS', and 'EE' together from 2004 to 18 May 2021. The terms which are used in the academic literature but not widely used by the public, such as EbA/EbM and BGI/GBI, do not register a trend due to insufficient search volume. The term NbS shows an increase in search volume since around 2014, reflecting an increasing public interest in the subject. The search volumes of GI, BI, and EE stopped growing since about ten years ago. GI has much larger search volumes than BI, EE, and NbS. The web volume for the term BI was much higher than NbS.

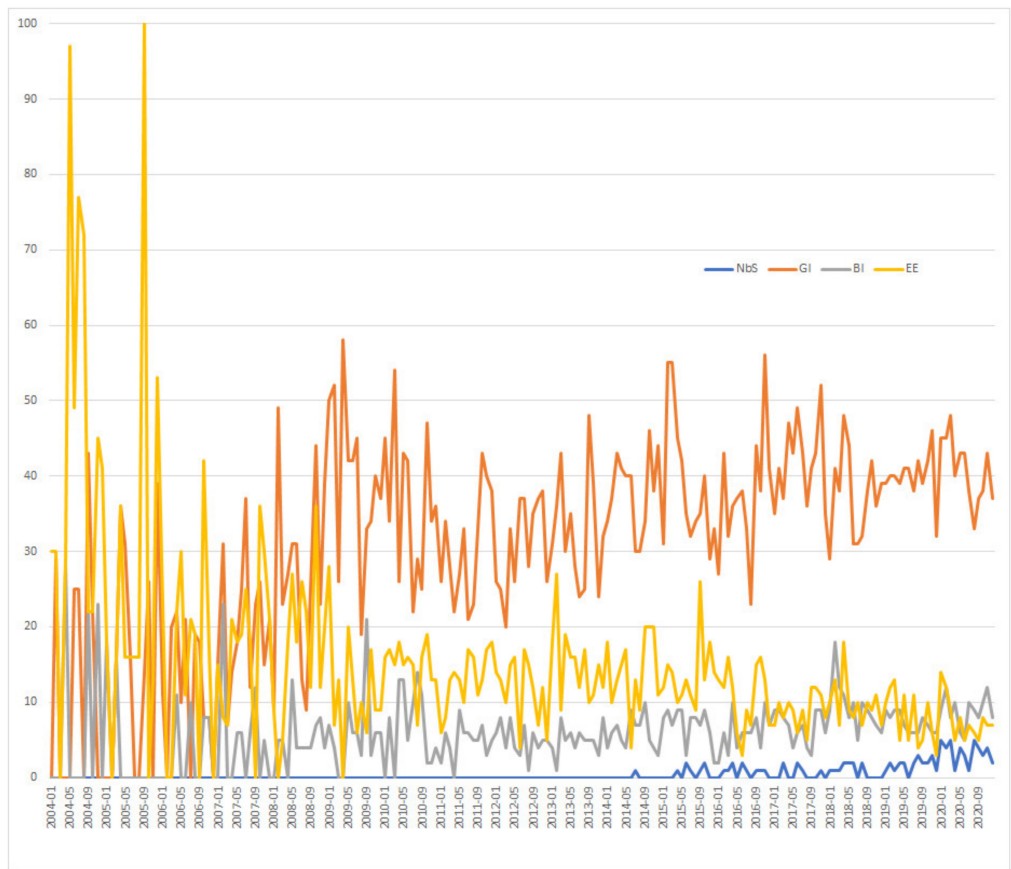

**Figure 4.** Trend in the terms 'NbS', 'GI', 'BI', and 'EE' over time. The *y*-axis is a relative volume expressed between 0 and 100, where the maximum search volume is set to 100.

In terms of geographical coverage, the interest levels of using these terms are different in different regions of the world within different contexts, partly due to the uptake in policy documents and social media. For instance, even though the concept of NbS has gained momentum and recognition in several international fora, including the UN Framework Convention on Climate Change (UNFCCC) [52], the Convention on Biological Diversity (CBD) [53], the Sendai Framework for Disaster Risk Reduction (DRR) [54], the World Economic Forum [55], and the UN Environment Assembly [56], among others, the concept of EbA/EbM is much more present there than NbS and some parties are still reluctant to introduce a new concept of NbS.

Figure 5 shows the interest by regions for these terms since 2004. Values are calculated as a value from 0% to 100%. Almost 100% of the searches in China were on EE. However, because Google has been banned by the Chinese government since 2010, this result largely reflected the searches in China between 2004 and 2010. In contrast, in France, searches on GI accounted for around 80% of all searches. Overall, we can see that the term NbS is most popular in European countries, such as the Netherlands, the UK, and Switzerland. The usage of the term NbS is also widespread in Singapore and Canada. The GI term is most used in France, the UK, and Ireland. The BI term is widespread in India, Australia, and Canada, followed by the UK and the USA. There is not enough data to show in which location the terms 'EbA/EbM' and 'BGI/GBI' were most popular during the timeframe of 2004–18 May 2021.

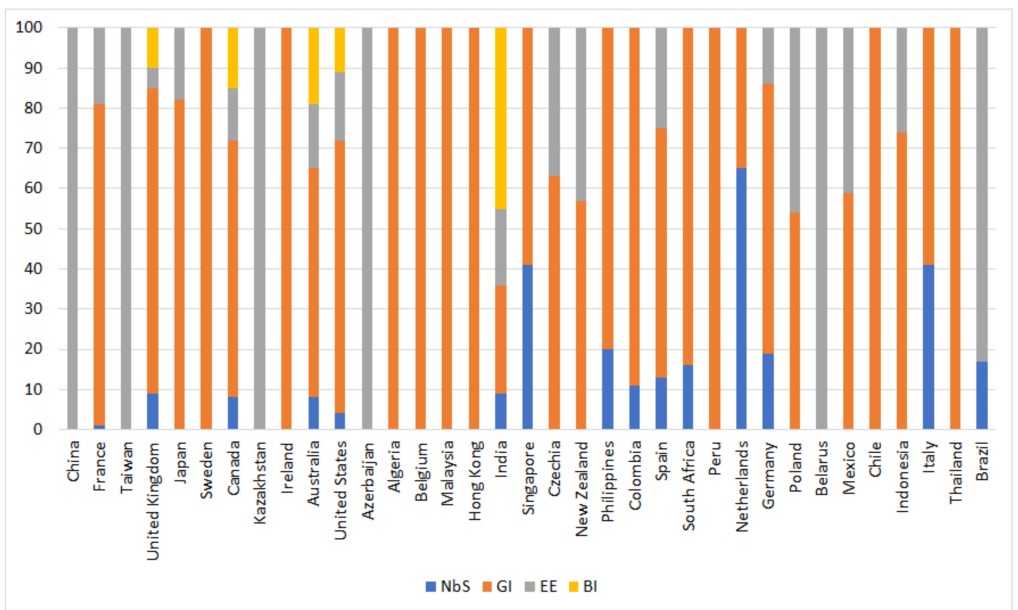

**Figure 5.** Interest by region of the terms 'NbS', 'GI', 'BI', and 'EE'.

## 3. The State-of-the-Art of the Nature-Based Solutions

*3.1. Global Frameworks of NbS Governance*

3.1.1. Nature-Based Solutions within International Intra-Governmental and Non-Governmental Governance Frameworks

The realization and the opportunity for broader use of NbS by incorporating nature into a range of sectoral and overarching strategies to meet societal challenges, at the global scale, has grown significantly in recent years. The increasing recognition of NbS's role in providing benefits to people is addressed in various international policy frameworks, such as the UN 2030 Agenda for Sustainable Development, the UN's Framework Convention on Climate Change (UNFCCC) and its Paris Agreement, the Ramsar Convention on Wetlands, the UN's Convention to Combat Desertification (UNCCD), and the Sendai Framework for DRR, etc.

According to IUCN, more than 130 countries have already included NbS actions (e.g., reforestation, GI, sustainable agriculture and aquaculture, and coastal protection) in their national climate plans under the Paris Agreement [18]. In the year of 2020, the IUCN and its Commission on Ecosystem Management (CEM) developed the Global Standard for NbS [18,36], which aimed to provide a foundation for the development of standards for implementing and upscaling NbS [10].

In addition to the EbA/EbM, the NbS have also been promoted in decisions of the UN CBD [57,58]. For example, in the year of 2019, in the Intergovernmental Science-Policy Platform on Biodiversity and Ecosystem Services (IPBES) Global Assessment report [59] and the UN CBD's Update of the Zero Draft of the Post-2020 Global Biodiversity Framework (GBF) in 2020 [60], the importance of recognizing and preserving nature's contribution to people as a key goal was addressed and NbS was seen as a key approach through which such goal can be achieved. Such an approach is also central to the CBD's mandate. Parties of the CBD promoted integration of NbS in the GBF as a pathway to achieving the 2030 action targets and ultimately the 2050 Vision [60].

Furthermore, the Sendai Framework for DRR called specifically for the development of standards for GI in order to stimulate investments in NbS [61]. Moreover, at the 2019 UN Climate Action Summit, NbS was one of the main topics discussed to combat climate change [62]. The role of strategic urban planning involving NbS is highlighted in both the EU-China and the EU-Latin American Partnership on Urbanization as well [63].

The UN New Urban Agenda makes specific reference to NbS for urban and territorial planning [64] and highlights the importance of biodiversity and the functioning

of ecosystems to maintain economic activities and the well-being of local communities. Considering NbS as solutions can support biodiversity conservation and the functioning of ecosystems, manage water-related risks, and transform natural capital into a source of green growth and sustainable development; in this context, NbS has the potential to contribute to the 2030 Agenda for Sustainable Development's targets and to help achieve the SDGs by delivering simultaneously various ES, and generating different social, economic and environmental co-benefits [10,65]. According to several studies on the relation of NbS to the different SDGs [66–69], NbS are directly relevant to SDG 2 (food security), 3 (health and well-being), 6 (clean water and sanitation), 10 (reduce inequality), 11 (sustainable cities and communities), 13 (climate change), 14 (conservation and sustainable use of oceans, seas, and marine resources), and 15 (protection, restoration, and promotion of sustainable use of terrestrial ecosystems)

### 3.1.2. Nature-Based Solutions within EU Research and Innovation Agenda

The EU has played a leading role in the international mainstreaming of policies that innovate with nature and contribute specifically to implementing the UN 2030 Agenda for SDGs [70], and supporting the development and implementation of NbS on climate adaptation and mitigation, disaster risk reduction, biodiversity protection, health, and well-being improvement. NbS are included in the key priorities of the EU R&I agenda. Since 2016, the EU has been supporting science-policy-business-society stakeholder dialogue platforms (such as ThinkNature) and promoting the market uptake of NbS. In 2017, as part of the Presidency of the Estonian Republic of the Council of the European Union, the "NbS: from innovation to common-use" conference was organized by the Ministry of the Environment of Estonia, focusing on policy and governance of NbS.

Through EC's Community Research and Development Information Service (CORDIS) [71], with the keywords 'Nature-based solutions', we found out that until 18 May 2021 (the day we implemented this search), 52 projects with a total budget of 435 million EUR have been funded by the EU R&I program since 2011 (Appendix A, Table A1). Table A1 provides an overview of the 52 projects which aim to create a community of practice/ecosystem on NbS, including major calls, funding scheme, project overall budget, lead country and partner countries, scope, challenges addressed, and type of NbS developed or implemented.

To summarize, there are seven types of NbS projects/actions funded by the EU R&I program (Table 2, Table A1; A detailed description of types of projects/actions can be found in the EC's General Annexes of the Main Work Program for Horizon 2020 [72], FP7 [73], and in the ERC Work Program [74]), including (i) coordination and support actions (CSA, 10 projects), such as "ThinkNature" and "NetworkNature" for multi-stakeholder dialogue platforms to promote NbS to societal challenges, "We Value Nature" and "MAIA" for mainstreaming natural capital in policies and in business decision-making, and "Eklipse" for focusing on method lies of knowledge learning mechanisms of biodiversity and ES; (ii) Research and innovation actions (RIA, 12 projects), e.g., projects "Nature4Cities" and "Naturvation" focusing on new governance, business, financing models, and economic assessment tools, project "NAIAD" for operationalizing insurance value of ecosystems, and projects "PONDERFUL", "DRYvER", "MaCoBioS", and "FutureMares" for inter-relations between climate change, biodiversity, and ES; (iii) Innovation actions (IA, 19 projects), e.g., projects on demonstrating innovation of NbS in cities (e.g., ConnectingNature, Grow Green, URBAN GreenUP, UNaLad, URBINAT, CLEVER Cities, proGIreg, EdiCitNet), projects on visionary and integrated solutions to improve well-being and health in cities (e.g., GO GREEN ROUTES, IN-HABIT, VARCITIES, EuPOLIS), and projects aiming to strengthen international cooperation on NbS for restoration and rehabilitation of urban ecosystems (e.g., CONEXUS, INTERLACE, REGREEN, CLEARING HOUSE); (iv) European Research Council-Networks (ERC-NET) cofunded actions (3 projects), such as "BiodivERsA" (78 sub-projects funded) on consolidating research about ES, biodiversity, and natural capital, and project "EN-SUGI" (15 sub-projects funded) on synthesizing the R&I expertise and

exploring innovative new solutions to the food-energy-water nexus challenge; (v) ERC-STG-Starting grants (1 project), e.g., the project Niche4NbS aiming to offer the capability to predict and plan the best NbS implementation; (vi) Marie Skłodowska-Curie Actions Individual Fellowships (MSCA-IF, 3 projects), such as the project aiming to investigate childhood heat-related health impacts and protective effects of urban natural environments (e.g., Green CURIOCITY); and (vii) collaborative projects (CP, 4 projects), such as the project 'GREEN SURGE' proving a sound evidence base for GI planning and implementation, and linking environmental, social, and economic services with local communities.

**Table 2.** The type/funding scheme of nature-based solutions projects funded by the EU R&I program (see Appendix A, Table A1).

| EU R&I Program | Type of NbS Projects and Their Definition | |
| --- | --- | --- |
| Horizon 2020 Program [72] | CSA—Coordination and support actions | Activities contribute to the objectives of Horizon Europe, which excludes R&I activities, except those carried out under the 'Widening participation and spreading excellence' component of the program. |
| | RIA—Research and innovation actions | Activities aim primarily to establish new knowledge or to explore the feasibility of a new or improved technology, product, process, service, or solution. |
| | IA—Innovation actions | Activities aim to directly produce plans and arrangements or designs for new, altered, or improved products, processes, or services. |
| | ERA-NET Co-funded actions | Action designed to support public–public partnerships (P2Ps), including joint programming initiatives between Member States, in their preparation, establishment of networking structures, design, implementation, and coordination of joint activities, as well as Union topping-up of a trans-national call for proposals. |
| | MSCA-IF—Marie Skłodowska-Curie Actions Individual Fellowship | MSCA-IF-GF—Global Fellowships: Support the international career of researchers by giving them the opportunity to conduct a research project in a host institution located in a Third Country. Mobility from an EU MS (European Member State)/AC (European-Associated Country) to any country for a 12–24-month fellowship + a return phase of 12 months in the EU. MSCA-IF-EF-SE—European Fellowships, Society, and Enterprise panel: The SE panel is a multidisciplinary panel of the European Fellowships, with an earmarked budget, dedicated to career opportunities for researchers seeking to work on research and innovation projects in an organization from the non-academic sector (e.g., businesses, civil society, and government bodies). MSCA-IF-EF-RI—European Fellowships, Reintegration panel: The Reintegration Panel is a multidisciplinary panel of the EF dedicated to researchers who wish to return and reintegrate in a longer-term research position in Europe. |
| EU FP7—Seventh Framework Program [73] | ERC-STG—European Research Council Starting Grant | ERC-STG are designed to support excellent Principal Investigators at the career stage at which they are starting their own independent research team or program. Principal Investigators must demonstrate the ground-breaking nature, ambition, and feasibility of their scientific proposal. |
| | CP—Collaborative Projects | CP-IP—Integrated Research Project (large research projects). CP—Collaborative project (generic): Activities are objective-driven research projects aiming at developing new knowledge, new technology, products, and that may include scientific coordination, demonstration activities, or sharing of common resources for research to improve European competitiveness or to address major societal needs. |

To summarize, the important principles in EU R&I-funded NbS-related projects (Appendix A, Table A1) are: (i) innovation in terms of social, governance, technical, regulatory, business, and finance aspects; and (ii) addressing multiple challenges and applying co-design to co-management processes. The NbS as a topic certainly will not end with EU Horizon 2020. Further topics focused on NbS implementation and innovation are also included in EU Green Deal Calls [30], and Horizon Europe calls on cluster 3 (NbS to enhance urban resilience and security (IA)), cluster 5 (Let nature do the job (RIA)), cluster 6 (Nature therapy (RIA), Socio-politics of NbS (RIA), Education for NbS (CSA), Economics of NbS (RIA), Network for nature (CSA), Agroecology and agroforestry (several topics)) [31], and other funding programs such as 'cohesion policy' funds [75], Invest EU [76], or LIFE program [77].

### 3.1.3. Nature-Based Solutions within Chinese Governmental Framework

In the field of NbS, China has also carried out various explorations on approaches and practices, such as the ecological conservation redline system [78], local government heads as river chiefs and lake chiefs [79], international coalition for green development on the belt and road [80], natural forest protection, afforestation, and sustainable forest management [81], urban ecological restoration [82], and so on.

In September 2019, during the UN Climate Action Summit, China and New Zealand jointly took the lead for a NbS Coalition in promoting the work on NbS and identified NbS as an important global action [62]. In fact, NbS is aligned with China's "ecological civilization thought", the "harmony between man and nature" as described in the report of the 19th National Congress of the Communist Party of China, "lucid waters and lush mountains are invaluable assets", and "mountains, rivers, forests, fields, lakes and grasses form a community of shared life" [83,84].

It is worth mentioning that the Chinese 'Sponge City' initiatives are also quite in line with the NbS concept. The Chinese central government introduced the "Sponge City" concept in 2013 as an approach to tackle many water-related problems, such as flooding and water pollution. A city built on water-centered eco-infrastructure that act like sponges to retain rainwater and make use of natural forces to accumulate, infiltrate, and purify rainwater is called a "Sponge City" [85–87], which is totally opposite to the conventional solution of grey infrastructure [86]. The goal of China's Sponge City initiative is that by 2030, 80% of urban areas should absorb and re-use at least 70% of rainwater [88]. Since 2015, NbS for ecological "Sponge Cities" have been implemented at different scales in different Chinese cities, and include NbS such as terracing the slopes, building retention ponds and ponding the ground, dyking and ponding the swamps, islanding the lakes, restoration of wetlands and the floodplain, and the re-naturalization of the river course [89]. More than 600 cities in China are required to meet the Sponge City goals in the next decade [88].

### 3.2. Implementation and Assessment of Nature-Based Solutions

### 3.2.1. Type of Nature-Based Solutions Implemented

There are different ways to group the NbS and type of the NbS with different classification criteria. Table 3 summarizes the existing types of NbS with classification criteria and key references.

**Table 3.** Type of nature-based solutions with their classification criteria and key references.

| Type of Nature-Based Solutions | Classification Criteria | Key References |
|---|---|---|
| (1) Solutions that involve making better use of existing natural or protected ecosystems (e.g., measures to increase fish stocks in an intact wetland to enhance food security).<br>(2) Solutions based on developing sustainable management protocols and procedures for managed or restored ecosystems (e.g., re-establishing traditional agro-forestry systems based on commercial tree species to support poverty alleviation).<br>(3) Solutions that involve creating new ecosystems (e.g., establishing green buildings, such as green walls and green roofs). | Degree of NbS interventions | [90,91] |
| (1) Better use of protected/natural ecosystems, e.g., protection and conservation strategies in terrestrial (e.g., Natura 2000—a network of nature protection areas in the territory of the European Union), marine (e.g., marine protected area), and coastal areas (e.g., mangroves) ecosystems.<br>(2) NbS for sustainability and multifunctionality of managed ecosystems, such as agricultural landscape management, coastal landscape management, extensive urban green space management, and monitoring.<br>(3) Design and management of new ecosystems, for example, intensive urban green space management, urban planning strategies, urban water management, ecological restoration of degraded terrestrial ecosystems, restoration and creation of semi-natural water bodies and hydrographic networks, and ecological restoration of degraded coastal and marine ecosystems. | The degree of intervention/level and type of engineering in many (sub)categories | [17] |
| (1) Greening interventions<br>(2) Public green space<br>(3) Vertical greening<br>(4) Green roofs<br>(5) Water-sensitive urban design measure<br>(6) River restoration<br>(7) Measure of bioengineering<br>(8) Other NbS | NbS planning and construction terminology | [92] |
| (1) Green NbS<br>(2) Blue NbS<br>(3) Hybrid NbS | Type of engineering, the type of ecosystem, and ecosystem functions level. | 52 EU-funded projects (Table A1) |

To summarize, at the engineering and ecosystem functions level, the NbS projects' websites and applications (Appendix A, Table A1) can be divided into three main approaches: (1) green (e.g., parks, forests), (2) blue (e.g., rivers, channels, lakes, ponds), and (3) hybrid (e.g., combining green/blue or blue/green and grey infrastructure approaches). Of the 52 projects websites/applications reviewed in Europe (Appendix A, Table A1), 22 were developed for or tested the green type of NbS, 18 were developed for or tested the blue type of NbS, and 33 were developed for or tested the hybrid type of NbS. Therefore, most sites/applications were focused on hybrid types of NbS.

### 3.2.2. Challenges Addressed by the Nature-Based Solutions and Their Key Performance Indicators for Impacts Assessment

NbS refers to the sustainable management and use of nature for tackling various challenges. In this study, such challenges are further categorized into three groups, including: (i) environmental challenges (e.g., climate change, urban sprawl, ecosystem degradation, soil sealing, landslides, heat stress, drought, storm surges, flooding, noise, environmental quality in public spaces, lack of green spaces, air pollution, biodiversity loss, water pollution, and food security) [93–101], (ii) economic challenges (e.g., lack of community capital, economic decline, employment issues, inefficient resource management, productivity, and affordability issues) [27,102], and (iii) social challenges (e.g., human health, quality of life, public participation, and equity) [27,102].

Moreover, there is a range of KPIs that have been developed to measure the impact of NbS or challenges addressed [92]. To acquire an overview of KPIs for NbS impact assessment, in addition to reviewing 52 EU-funded projects, for key relevant publications, we used the keywords 'nature-based solutions' and 'impact assessment' in Google Scholar and reviewed 33 articles in total (see key references column in Appendix B, Table A2). Appendix B, Table A2 provides a brief overview of existing indicators that have been developed and methodologies used to assess various NbS's impacts on the environment, economy, and society, respectively.

Of the 52 projects' websites/applications reviewed in Europe (Appendix A, Table A1), a majority of the projects focused on challenges related to urbanization, climate change and its implications for the environment, human health, and well-being. To summarize, the main challenges addressed by these 52 projects are climate change, air pollution, greenhouse gases (GHG) emissions, biodiversity loss, ES degradation, marine coastal ecosystem degradation, drying rivers, flooding, soil erosion, landslides, drought, heat stress, drinking water consumption, energy consumption, lack of adequate GI, lack of quality greenspaces, limited natural ecosystems, landscape fragmentation, and urban sprawl. Accordingly, the key impacts assessed by these projects are climate resilience and mitigation, biodiversity enhancement, microclimate regulation and air quality improvement, flood mitigation and coastal resilience, improving water quality and waterbody conditions, sustainable communities, innovative governance, and business models. For completed projects, most focus solely on environmental benefits delivered by NbS, and only very few have looked at economic benefits of NbS and their contribution to reducing social injustice and improving social capacity building and cohesion (ECLIPSE, ThinkNature, and NAIAD). However, several ongoing and newly funded projects (e.g., REGREEN, UrbiNAt, EdiCitNet, ProGIreg, CleverCities, Grow Green, Connecting Nature, NATURVATION, and Nature4Cities) have highlighted the necessity to assess NbS's impact on social justice and social cohesion, new economic opportunities, and green jobs.

### 3.2.3. Nature-Based Solutions Data and Metadata

As of July 2021, there are several databases and platforms focusing on NbS and their related environmental topics, which allow the users to share knowledge and experience, and combine and use data and information at national, regional, and global levels. For example, the ThinkNature platform [17,103], Oppla—EU Repository of NbS [19], NbS initiative [104], and Urban Nature Atlas [105] are platforms on diverse environmental challenges directly addressed by NbS, while the European Environmental Agency (EEA)

Data and Maps [106], EEA Climate-ADAPT [107], EU Joint Research Centre (JRC) Data Catalogue [108], Copernicus—Europe's Eye on Earth [109], EU Biodiversity Information System [110], EU water information system [111], and EU forest information system [112] are databases and platforms which have systematically collected diverse data, including data on specific natural hazards, that may not be directly related to NbS, but can be useful when designing and implementing NbS, for example, certain type of hazards can be mitigated or resolved by NbS. Table 4 summarizes the selected NbS databases and platforms with their brands and brief descriptions.

**Table 4.** Selected nature-based solutions databases and platforms.

| Database/Platform | Brand | Description |
|---|---|---|
| ThinkNature | A multi-stakeholder communication platform [103] | Supporting the understanding and promotion of NbS. |
| Oppla | Open platform that hosts the EU Repository of NbS [19] | The EU Repository of NbS, which provides a knowledge marketplace, where the latest thinking on natural capital, ecosystem services, and NbS is brought together. |
| NbS initiative | Interdisciplinary program [104] | NbS initiative is an interdisciplinary program, seeking to apply impactful research to shape policy and practice on NbS through research, teaching, and engagement with policymakers and practitioners. |
| Urban Nature Atlas | Atlas [105] | It contains 1000 examples of NbS from across 100 European cities. |
| Climate-ADAPT | European Climate Adaptation Platform [107] | It supports Europe in adapting to climate change and in helping users to access and share data and information on:<br><br>• Expected climate change in Europe<br>• Current and future vulnerability of regions and sectors<br>• EU, national, and transnational adaptation strategies, and actions<br>• Adaptation case studies and potential adaptation options including NbS<br>• Tools that support adaptation planning |
| EEA Data and Maps | EEA Data and Maps provides access to datasets used in EEA periodical reports [106] | Thematic topics of EEA data and maps:<br><br>• Agriculture<br>• Air pollution<br>• Biodiversity—Ecosystems<br>• Chemicals<br>• Climate change adaptation<br>• Climate change mitigation<br>• Default<br>• Energy<br>• Environment and health<br>• Environmental technology<br>• Industry<br>• Land use<br>• Marine<br>• Policy instruments<br>• Resource efficiency and waste<br>• Soil<br>• Specific regions<br>• Sustainability transitions<br>• Transport<br>• Water and marine environment |
| JRC Data Catalogue | The JRC Data Catalogue provides access to the multidisciplinary data produced and maintained by the JRC [108] | Thematic scope of JRC data catalogue:<br><br>• Environment and climate change<br>• Agriculture and food Security<br>• Economic and monetary union<br>• Energy and transport<br>• Health and consumer protection<br>• Information society<br>• Innovation and growth<br>• Nuclear safety and security<br>• Safety and security<br>• Standards |

**Table 4.** *Cont.*

| Database/Platform | Brand | Description |
|---|---|---|
| Copernicus | The European Earth Observation Program [109] | It is the European system for monitoring the Earth and is coordinated and managed by the EC. Its services address six thematic areas:<br><br>• Land<br>• Marine<br>• Atmosphere<br>• Climate change<br>• Emergency management<br>• Security |
| BISE—Biodiversity Information System for Europe | The source of data and information on biodiversity in Europe [110] | It is a single-entry point for data and information on biodiversity supporting the implementation of the EU strategy and the Aichi targets in Europe. |
| WISE—Water Information System for Europe | European information gateway to water issues [111] | It is a joint initiative from the EC (DG Environment, JRC, and Eurostat) and the EEA to modernize and streamline the collection and dissemination of information related to European water policy. |
| FISE—Forest information system for Europe | The entry point for sharing information with the forest community on Europe's forest environment, its state, and development [112] | It brings data, information, and knowledge gathered or derived through key forest-related policy drivers. |

### 3.2.4. Stakeholders' and Citizens' Prioritization for Nature-Based Solutions Implementation

Stakeholder and citizen participation and collaboration in NbS are increasingly recognized as promising [113]. However, there are different prioritizations for different beneficiaries. For example, for general citizens, NbS need to be aesthetically appealing [114]. For urban planners, they need to have an open approach to collaborative governance of NbS that allows learning with and about new attractive designs, perceptions, and images of nature from different urban actors, and allows forming new institutions for operating and maintaining NbS to ensure inclusivity, livability, and resilience [114].

Of the 52 projects websites/applications reviewed in Europe (Appendix A, Table A1), a majority of NbS were typically implemented in cities on a small scale to target specific types of challenges, particularly linking to urbanization and climate change and its implications for environment and society, such as projects on water and climate resilience (Connecting Nature, Grow Green, Urban GreenUP, UNaLab), inclusive urban regeneration (CLEVER Cities, EdiCitNet, ProgIreg, URBINAT), decarbonization and air quality (DivAirCity, Upsurge, JUSTNature), improving health and well-being (EUPOLIS, GO GREEN ROUTES, IN-HABIT, VARCITIES), governance-business-financing models and economic impact assessment tools (NATURVATION, Nature4cities), restoration and rehabilitation of urban ecosystems (CLEARING HOUSE, REGREEN, CONEXUS, INTERLACE), inter-relationship between climate change, biodiversity, and ES (DRYvER, FutureMARES, MaCoBioS, PONDERFUL), and insurance value of ecosystems (NAIAD). Only a small number of projects (OPERANDUM, PHUSICOS, RECONECT) implemented NbS in rural and mountainous areas at a relatively large scale and targeted the challenges related to hydro-meteorological hazards, particularly, e.g., extreme weather events, flooding, erosion, landslides, and drought. This may be due to the challenges of the NbS implementation, cost, and governance at large scales. However, there is a lot of work going on at the moment under the umbrella of re-wilding [115]. Re-wilding is now a widespread approach across Europe that has some different aims, but with some overlap through common aims with NbS in terms of promoting better ES provision, for instance re-wilding of certain mountainous areas can support soil stability and prevent excessive runoff, leading to reduced flood impacts and landslide risk [116].

## 4. Discussion

### 4.1. Nature-Based Solutions as a Strong Concept That Is Becoming Recognized and Accepted

NbS is a term that is defined and used differently by a number of stakeholders. Until now, the IUCN and the EC have developed their own definitions of NbS, which while broadly similar have different focuses, for example, they share the overall goal of addressing major societal challenges through the effective use of ecosystem and ES (See Section 2.1). The IUCN's definition was developed from a global perspective and emphasizes the need for a well-managed or restored ecosystem to be at the heart of any NbS, while the EC definition places more emphasis on applying solutions that not only use nature but are also inspired and supported by nature [117]. The IUCN [27] (p. 10) proposed to consider "NbS as an umbrella concept, which covered ecosystem-based management and issue-specific ecosystem related approaches (e.g., EbA/EbM, ecosystem-based disaster risk reduction), infrastructure-related approaches (e.g., GI, BI, BGI/GBI), and ecosystem protection approaches (e.g., EbA)". The EC suggested that the "NbS builds on and supports other closely related concepts, such as the EE, ES, EbA/EbM, GI and BI" [9] (p. 24). Both the IUCN and the EC emphasize NbS as concrete actions that cover a range of ecosystem-related approaches to solve problems at a local and regional scale [118].

Although there is no unified definition of NbS, the concept of NbS links the multiple potential positive outcomes for society. In fact, it is not a problem that there is a lack of a unified definition of NbS. The existing concepts of NbS cover a broad range of aspects dealing with the challenges of our time, e.g., biodiversity loss, climate crisis, and the need for building resilient futures for societies in urban, rural, and wild landscapes for healthy people and a healthy nature. NbS could as such be considered as an overarching concept for all related terms, which is in line with Lafortezza et al. [119] and Davies et al. [120] addressment in a European context. In addition, from an impact perspective, the NbS concept can be seen to encompass existing concepts such as EbA/EbM, as proposed by Rizvi et al. [121]. Furthermore, NbS can be used to build GI, BI, and BGI/GBI, but GI, BI, and BGI/GBI can also be part of a broader NbS infrastructure, as addressed by Balian et al. [122] (See Section 2.2).

The increasing use of the NbS concept in the literature shows the growing interest of the scientific community in using NbS as an overarching framework, on how to use nature to address multiple challenges and foster sustainability (See Section 2.3). On the other hand, globally, of the NbS concept remains less used in non-scientific discourses, but still gaining more and more interest since 2016, when the term 'NbS' was coined by the EC (See Section 2.4). With a disparate geographical anchor, NbS is in high usage, especially in some Western European countries. These trends of non-scientific use of NbS are also aligned with the scientific mapping of NbS in urbanism performed by Li et al. [123]. NbS is much less common than the GI concept in general when considering non-scientific search engines such as Google Trends (see Section 2.4), which is not very surprising given that NbS is a relatively new phrase. To the best of our knowledge, no study has been carried out on analyzing the emergence of NbS and the related concepts over time in broader discourses towards more layman outlets, and the geographical coverage for the different NbS-related concepts. Such type of analysis is quite important in terms of awareness raising of NbS concepts, broad participation in developing a NbS, successful implementation of NbS, etc., which are aligned well with the IUCN's NbS core principles, in particular principle 3 (NbS are determined by site-specific natural and culture contexts that include traditional, local, and scientific knowledge), and 4 (NbS produce society benefits in a fair and equitable way in a manner that promotes transparency and broad participation) [10].

### 4.2. Need for Good Practices of Implementation and Stronger Evidence of the Benefits of Nature-Based Solutions to Tackle Environment, Health, and Well-Being Challenges

NbS can be applied strategically and equitably to help societies address a variety of climatic and non-climatic challenges. Based upon the review of 52 NbS projects' websites/applications (Appendix A, Table A1) and some relevant publications, the key areas

for NbS practices and evidence developed are biodiversity, climate change mitigation and adaptation (including flooding), water quality, air quality and microclimate, sustainable communities, innovative governance and business models, and market challenges and solutions [124]. NbS can provide low-risk, low-maintenance, and low-cost solutions to climate change-related hazards and impacts [125,126], but there is still a lack of understanding on how best to implement them in practice, in an up-scaled manner, and on how to strengthen integration within institutions. Several studies identified the challenges to implement NbS in the future planning and management of green/blue landscape and to include institutional changes (e.g., policy, governance, and culture) for future refinements of the NbS concept and its applications in both rural and urban landscapes [9,127,128]. The EU-funded project ThinkNature addressed that NbS are energy- and resource-efficient, and resilient to change, but to be successful, they must be adapted to local conditions [17] (Appendix A, Table A1), which is also recognized by the IUCN[129,130].

The IUCN addressed three major challenges for NbS implementation and its impact evaluation [130]. First, challenges in measuring or predicting the effectiveness of NbS lead to high uncertainty about their cost-effectiveness compared to alternatives, which is also well-reflected by very few projects funded by the EC on assessing cost-effectiveness and cost-economic benefits (e.g., ECLIPSE, ThinkNature, and NAIAD, see Appendix A, Table A1). Second, poor financial models and flawed approaches to economic appraisal led to underinvestment in NbS. Third, inflexible and highly sectorized forms of governance hinder uptake of NbS, where grey and engineered interventions are still the default approach for climate adaptation and mitigation challenges. Such challenges on participatory planning and governance are also highlighted by several EU-funded NbS projects, such as ThinkNature, CONNECTING NATURE, UNaLab, URBAN GreenUP, and CLEVER Cities [92] (see Appendix A, Table A1).

NbS has wide applications in environment, society, and economy, but in fact, it is quite difficult to assess its impact on environmental–social–economic benefits together, which is well-reflected by the few studies that have assessed or are aiming to assess simultaneously social, economic, and environmental benefits (i.e., OpenNESS, EKLIPSE, GREEN SURGE) [69]. Within the 52 projects and some other relevant publications reviewed, the environmental research fields dominate use and interpretation of the NbS in practices, which is aligned well with the results highlighted by Hanson et al. in their study on 'Working on the boundaries—how does science use and interpret the NbS concept' [69]. Therefore, there is a need for stronger evidence of the benefits of NbS to tackle environmental, health, and well-being challenges simultaneously. To do so, it is important to understand the value and limits of NbS, such as NbS's reliability and its cost-effectiveness by comparing it to grey-engineered interventions, and NbS's resilience to climate change and its co-benefits and trade-offs. For example, benefits in one challenge area (e.g., green infrastructure) can have co-benefits, costs, or neutral effects in other challenge areas (e.g., improvement of place attractiveness, health and well-being, creation of green jobs) [131]. Trade-offs can arise if climate mitigation policy encourages NbS with low biodiversity value, such as afforestation with non-native monocultures, which can result in maladaptation, particularly in the regions where biodiversity-based resilience and multi-functional landscapes are the key [129]. Therefore, the challenges addressed by the NbS and its environmental–social–economic benefits will not be realized unless they are implemented within a system-thinking framework that accounts for multiple ES and recognizes trade-offs and synergies among them from the perspective of different stakeholders [129].

Moreover, it is challenging to assess NbS using a multitude of sensors and data sources, including remotely sensed images (e.g., high-resolution satellite sensors, field sensors, and airborne LiDAR) and field data [132], and scale-up NbS benefits to the global level and provide evidence metrics or indicators that managers and policymakers can easily access and use [119]. Hunt et al. [133] addressed the need for stronger evidence of the benefits of nature to tackle the problem of dementia. Lafortezza et al. [119] addressed the challenges to understand the linkages between NbS and associated ES within the four main categories

of provisioning, regulating, cultural, and supporting across different scales (e.g., from the "core" urban area to the wider peri-urban landscape). NbS benefits will not be released unless they are implemented within a systems-thinking framework that fully accounts for their potential to support multiple ES and the trade-offs among them [68,130].

*4.3. Need for Sustainable Design of Nature-Based Solutions*

There are several aspects that need to be considered for sustainable design and co-design of NbS. Based upon the review of 52 EU-funded projects and some relevant publications (Sections 3.2.1, 3.2.2 and 3.2.4, Appendix A, Table A1), the following aspects of NbS shall be addressed. First, the scope and nature of the problems that need to be solved shall be defined [7,27,134], including many aspects, such as: (i) Are these problems short-lived, long-lasting, or permanent? (ii) Who are the stakeholders? and (iii) What are the potential difficulties in the solution?

Secondly, nature's boundaries, i.e., the aspects related to the biological and abiotic composition of the ecosystem, should be considered [7,27,134], such as: (i) Can an ecosystem be considered as a whole, including water, material circulation, and energy flow? (ii) Can landscape and urban environment with artificial structure and human beings be included? and (iii) Can an ecosystem's self-sustaining potential be regarded as an aspect of natural attributes and sustainability? Most NbS have an impact on the ecosystem composition, functionalities, and features to some extent, e.g., while selecting certain ES and certain species combinations. This also means that there will be trade-offs that must be evaluated. This is a matter of multiple complexity. For example, restoration of a wetland for flood control may have a positive impact on a variety of ES, such as climate regulation, water purification, and provision of habitat and ecotourism. However, at the same time, it will have a negative impact on local agricultural production, which is a trade-off [134].

Third, the participation of multiple stakeholders (e.g., financiers, planners, designers, and innovators) and their creative dialogue shall be ensured, as addressed by EU-funded projects "ReNature", "NBS2017", and "CLEVER Cities" (Appendix A, Table A1). To realize the multi-stakeholder participation, the following questions shall be discussed, including: (i) Do these solutions rely on technological or physical innovation? and (ii) How to ensure social cohesion and fairness and how to judge the fairness? In most cases, key decisions about NbS design, cost, location, size, and the level of management intensity will involve a wide range of stakeholders, who may have different ideas and ways of managing these issues [7,134]. Then, it is necessary to ensure the participation of multiple stakeholders, since their views, considerations, and knowledge can provide information for planning and improve the planning. The participation of stakeholders will increase the adequacy and legitimacy of a NbS.

Fourth, NbS projects also need to be integrated with multidisciplinary and interdisciplinary fields, as addressed by the project "ReNature" (Appendix A, Table A1). In fact, in many restoration and rehabilitation projects, close cooperation is needed among ecological science, engineering, and social science, to jointly deal with the problem of how to provide ES [7,134]. With the development of NbS, there may be more demands for interdisciplinary and multidisciplinary integration to foster trans-disciplinarity. Moreover, as an ambitious policy instrument, NbS needs strategic design. Large-scale NbS will require several years or even decades of management. If the political situation or political agenda changes too often, it is difficult to ensure the effective implementation of long-term NbS. Therefore, it is necessary to maintain the stability of the policy, which will require a broad consensus among political parties on the NbS policy.

## 5. Conclusions

This paper analyzed NbS and the related concepts, including the Chinese Sponge City concept, investigated the global frameworks of NbS governance, and addressed NbS's benefits, challenges, and development needs for its implementation and assessment.

Based on the 52 NbS projects' review and synthesis presented, and the discussion above, we conclude the following:

(1) NbS and its related concepts: There is a common understanding and increasing recognition that NbS are valid solutions for securing ES and improving environmental quality while bringing important health, and well-being benefits. NbS has been broadly promoted by the IUCN and the EC as living solutions and actions to tackle various societal challenges. The concept of NbS is aligned very well with the concepts of EE, BI, GI, GBI/BGI, EbA/EbM, and the Chinese 'Sponge City' concept.

(2) Emergence and frequency of NbS over time and its geographical coverage: There is a growing interest of the scientific community in using NbS. NbS concept remains less used in non-scientific discourses, but they are still gaining more and more interest. There is a disparate geographical anchor of NbS. NbS is mainly used in Western European countries, such as the Netherlands, the UK, and Switzerland.

(3) NbS categorization: The challenges addressed by the NbS and the KPIs used to assess NbS impacts can be categorized in different ways. In this study, we grouped the challenges addressed by the NbS into environmental, economic, and social aspects; accordingly, we grouped the NbS impacts and the KPIs to assess the impact into environmental, economic, and social impacts as well. There are also different ways to categorize the type of NbS. In this study, we divided NbS into three types with different classification criteria, i.e., green, blue, and hybrid NbS. Most NbS implemented in Europe were focused on hybrid types of NbS.

(4) NbS implementation and assessment: The majority of NbS implemented in Europe are at the city level, and on relatively small scales to target challenges particularly linked to urbanization and climate change and their implications for the environment and society. Only a small number of NbS are implemented in rural and mountainous areas, at relatively large scales, and target the challenges related to hydro-meteorological hazards. Additionally, only a few projects have looked at the economic benefits of NbS and their contribution to reducing social injustice and improving social capacity building and cohesion. There are concerns over NbS's reliability and cost-effectiveness compared to grey-engineered alternatives. There is a need for more research on beneficiaries in NbS evaluation, particularly economic benefits and contribution to social justice, social cohesion, new economic opportunities, and green jobs.

(5) NbS data and metadata platforms: There are several existing NbS-related data and metadata platforms, including platforms in which environmental challenges can be addressed directly by NbS, such as ThinkNature, Oppla, NbS initiative, and Urban Nature. There are also platforms that have collected diverse data, which may not be directly related to NbS, but can be useful when designing and implementing NbS, such as EEA Data and Maps, EEA Climate-ADAPT, JRC Data Catalogue, EU Copernicus, EU Biodiversity Information System, EU water information system, and EU forest information system.

(6) NbS benefits and challenges of implementation and assessment: There is a great amount of evidence for NbS benefits for restoration and rehabilitation of ecosystems, carbon neutrality, and improved environmental quality, eventually improving health and well-being. However, the mechanism of NbS provision of the intended benefits, especially of combined multiple benefits of one and several NbS, still need to be better understood; especially, co-benefits, synergies, and trade-offs have not been systematically measured in diverse structures, configuration, and scale.

There is a lack of recommendations of optimal NbS and appropriate typologies fitting to different contexts in terms of different climatic, environmental, and social-economic conditions and different urban design. Although tools, models, design guidelines, standards, and protocols exist, there is still a need for an integrated and system-thinking framework for NbS implementation and impact evaluation, that integrates NbS into local policy frameworks, socio-economic transition pathways, and spatial planning. Filling these knowledge and evidence gaps will make strong cases for wide deployment and success-

ful implementation of NbS. NbS also require co-creation and co-management settings to connect with urban social innovation, and a collaborative approach to their planning and implementation. It is important to see the innovative forces at work when planning NbS and the need to bring the scientific community, the private sector, and the policymakers together. The financial and governance challenges are the major barriers to implementing NbS at scale. Reform in governments is required to allow desilofication and more flexible urban governance structures and to support collaborative bottom-up processes, such as grass root and civil society initiatives. For businesses, it is necessary to make the financial case for NbS, synthesizing the existing practices on sustainable and innovative financing of NbS, bringing actors of social innovations together, and developing promotional strategies and business models.

**Author Contributions:** H.-Y.L. initiated the study, undertook the review of NbS projects and other relevant publications, wrote the draft, and led the revisions of the manuscript; M.J. and X.C. wrote part of the manuscript, contributed to the refinement of the core idea of the study, and participated in the revisions of the manuscript. All authors have read and agreed to the published version of the manuscript.

**Funding:** This research received no external funding.

**Institutional Review Board Statement:** Not applicable.

**Informed Consent Statement:** Not applicable.

**Data Availability Statement:** Not applicable.

**Acknowledgments:** Open-access publication was funded by NILU—Norwegian Institute for Air Research. Liu received funding from the NILU-NIVA SIS project on urban sustainable development, EEA ETC/CME and ETC/ATNI tasks on urban environmental sustainability, and NILU internal project on nature-based solutions for improving environmental quality, health, and well-being. We would like to thank the anonymous referees for their very valuable comments. Special thanks to Paul D. Hamer, at the Urban Environment and Industry Department (NILU), for helping us with the language. Any remaining errors are the responsibility of the authors.

**Conflicts of Interest:** The authors declare no conflict of interest.

# Appendix A

**Table A1.** EU FP7 and H2020 calls on NbS, funded projects, and total findings from the EU R&I program and other funding sources (CSA: Coordination and support actions; IA: Innovation action; PP: Public procurement; RIA: Research and innovation action; CP: Collaborative project; IP: Integrating project; CP-IP: CP-IP—Large-scale integrating project; NA: Not applicable/Not available).

| | Type of Call/Funding Scheme | Funded Projects/Duration/Scope/Web Portal | Budget (€) | Lead Country/Partners Countries/No. of Partners | Study Area/Scale | Challenged Addressed | NbS-Related Concept | Type of NbS Developed/Tested |
|---|---|---|---|---|---|---|---|---|
| ENV.2011.2.1.5-1—Sustainable and Resilient Green Cities | CP-IP | TURAS—Urban resilience and sustainability (2011–2016): test the feasibility of urban sustainable transition approaches in selected case study neighborhoods to enable adaptive governance, collaborative decision-making, and behavioral change towards resilient and sustainable European cities. https://cordis.europa.eu/project/id/282834 (accessed on 28 September 2021)http://r1.zotoi.com (accessed on 28 June 2021) | 8,869,491 | IE/UK, NL, RS, DK, DE, BG, SI, IT, BE, ES, IE/33 partners (11 countries) | Europe | Climate change Natural resource shortage Unprecedented urban growth | GI | Green (e.g., renaturing the city, green roofs, green living room, pocket park, agriculture land, landscape park, urban gardens) |
| ENV.2012.6.2-1—Exploration of the operational potential of the concepts of ecosystem services and natural capital to systematically inform sustainable land, water, and urban management | CP | OPERAS—Ecosystem science for policy & practice (2012–2017): apply the ecosystem services and natural capital concept into practice. https://cordis.europa.eu/project/id/308393 (accessed on 28 June 2021) http://operas-project.eu (accessed on 28 June 2021) | 11,459,749 | UK/NL, DE, SE, FI, BE, UK, RO, IE, FR, CH, BG, ES, PT, ES, ID/29 partners (15 countries) | Europe and Indonesia | Climate change Ecosystem degradation | EbA/EbM | Green (e.g., green space, Montado, Circum-Mediterranean agricultural land) Blue (e.g., coastal environment, islands) Hybrid (e.g., coastal 'cultural ecosystem services') |

Table A1. *Cont.*

| Type of Call/Funding Scheme | Funded Projects/Duration/Scope/Web Portal | Budget (€) | Lead Country/Partners Countries/No. of Partners | Study Area/Scale | Challenged Addressed | NbS-Related Concept | Type of NbS Developed/Tested |
|---|---|---|---|---|---|---|---|
| | OPENNESS—Operationalization of natural capital and ecosystem services (2012–2017): develop innovative and practical ways of applying ecosystem services and natural accounting in land, water, and urban management, and to identify how, where, and when the concepts of ecosystem services and natural accounting can most effectively be applied to solve problems. https://cordis.europa.eu/project/id/308428 (accessed on 29 June 2021) http://www.openness-project.eu (accessed on 28 June 2021) | 11,488,446 | FI/RO, DE, NL, UK, NO, BE, FI, FR, HU, SK, PT, ES, DK, IT, AT, BR, AR, KE, IN/39 partners (19 countries) | Europe, India, Brazil, Argentina, Kenya | Climate change Ecosystem degradation | EbA/EbM | Green (e.g., woodlands, grasslands, and farmlands) Blue (e.g., freshwater bodies, coastal zones, woodlands) Hybrid (e.g., a range of social-ecological systems including river basin, coastal zone, urban and regional planning, and their interfaces) |
| ENV.2013.6.2-5—Urban biodiversity and green infrastructure | GREEN SURGE—Green Infrastructure and urban biodiversity for sustainable urban development and the green economy (2013–2017): identify, develop, and test ways of connecting green spaces, biodiversity, people, and the green economy. https://cordis.europa.eu/project/id/603567 (accessed on 28 September 2021) https://ign.ku.dk/english/green-surge (accessed on 30 June 2021) | CP     7,189,726 | DK/FI, DE, NL, SE, UK, HU, IT, PO, PT, SI, DK/24 partners (11 countries) | Europe | Land use conflicts Climate change Human health and well-being issues | GBI/BGI | Green (e.g., city parks, green walls, rooftop gardens, urban forests, allotment gardens) Blue (e.g., lakers, rivers) |

Table A1. *Cont.*

| | Type of Call/Funding Scheme | Funded Projects/Duration/Scope/Web Portal | Budget (€) | Lead Country/Partners Countries/No. of Partners | Study Area/Scale | Challenged Addressed | NbS-Related Concept | Type of NbS Developed/Tested |
|---|---|---|---|---|---|---|---|---|
| SC5-10b-2014—Structuring research on soil, land-use, and land management in Europe | CSA | INSPIRATION—Integrated spatial planning, land use and soil management research action (2015–2018): establish and promote the adoption of the knowledge creation, transfer, and implementation agenda for land use, land-use changes, and soil management. https://cordis.europa.eu/project/id/642372 (accessed on 28 July 2021) | 2,812, 585 | DE/FR, PO, CZ, PT, BE, IT, SI, CH, DE, UK, ES, NL, SK, AT, RO, FI/21 partners (16 countries) | Europe | Land use changes and soil management | EbA/EbM | NA |
| SC5-09-2014—Consolidating the European Research Area on biodiversity and ecosystem services | ERA-NET Co-funded | BiodivERsA3—Consolidating the European research area on biodiversity and ecosystem services (78 projects) (2015–2020): provide policymakers and other stakeholders with adequate knowledge, tools, and practical solutions to address biodiversity and ecosystem degradation. https://cordis.europa.eu/project/id/642420 (accessed on 28 September 2021)http://www.biodiversa.org (accessed on 1 July 2021) https://www.era-learn.eu/network-information/networks/biodiversa3 (accessed on 1 July 2021) | 38,974,333 | FR/AT, BE, BG, EE, FR, NC, DE, HU, LT, NL, NO, PO, PT, RO, ES, SE, CH, TR, UK, SK, FI, IE, IL/37 partners (24 countries) | Europe | Climate change Loss of biodiversity Degradation of ecosystems | EbA/EbM, BGI/GBI | Green (e.g., Beech forests, grassland, Congo Basin forests, heathlands) Blue (e.g., Marine Environments Hybrid (e.g., peatland; terrestrial, freshwater, and marine systems and its interactions) |

**Table A1.** *Cont.*

| | Type of Call/Funding Scheme | Funded Projects/Duration/Scope/Web Portal | Budget (€) | Lead Country/Partners Countries/No. of Partners | Study Area/Scale | Challenged Addressed | NbS-Related Concept | Type of NbS Developed/Tested |
|---|---|---|---|---|---|---|---|---|
| SC5-10b-2014—Structuring research on soil, land-use, and land management in Europe | CSA | INSPIRATION—Integrated spatial planning, land use, and soil management research action (2015–2018): adopt a funder and end-user demand-driven approach to establish and promote the adoption of the knowledge creation, transfer, and implementation agenda for land use, land-use changes, and soil management. https://cordis.europa.eu/project/id/642372 (accessed on 28 September 2021) http://www.inspiration-h2020.eu (accessed on 19 July 2021) | 2,812,586 | DE/FR, PO, CZ, PT, BE, IT, SI, CH, DE, UK, ES, NL, SK, AT, FI/21 partners (15 countries) | Europe | Soil, land use related challenges (e.g., climate change, depletion of natural resources and loss of biodiversity) | EbA/EbM | NA |
| SC5-10a-2014—Enhancing mapping ecosystems and their services | CSA | ESMERALDA—Enhancing ecosystem services mapping for policy and decision making (2015–2018): deliver a flexible methodology to provide the building blocks for pan-European and regional assessments in relation to the requirements for planning, agriculture, climate, water, and nature policy.https://cordis.europa.eu/project/id/642007 (accessed on 2 July 2021) http://esmeralda-project.eu (accessed on 2 July 2021) | 3,133,306 | DE/FI, ES, UK, IT, BG, NL, BE, CZ, CH, LV, HU, PT, RO, AT, PO, FR, MT, DK, SE, IE, NO, IL, SK, EE, LT, EL, CY, SI, HR, LU/39 partners (31 countries) | Europe | Loss of biodiversity Loss of ecosystem services | GI, Ecosystem services | NA |

**Table A1.** *Cont.*

| | Type of Call/Funding Scheme | Funded Projects/Duration/Scope/Web Portal | Budget (€) | Lead Country/Partners Countries/No. of Partners | Study Area/Scale | Challenged Addressed | NbS-Related Concept | Type of NbS Developed/Tested |
|---|---|---|---|---|---|---|---|---|
| SC5-10c-2015—An EU support mechanism for evidence-based policy on biodiversity & ecosystems services | CSA | EKLIPSE—Establishing a European knowledge and learning mechanism to improve the policy-science-society interface on biodiversity and ecosystem services (2016–2020): establish an open support mechanism at EU for evidence-based policy on biodiversity and ecosystems services. https://cordis.europa.eu/project/id/690474 (accessed on 28 September 2021) http://www.eklipse-mechanism.eu (accessed on 1 July 2021) | 3,117,272 | DE/UK, FI, BE, DE, FR, HU, RO, PT/12 partners (8 countries) | Europe | Biodiversity loss and ecosystem services degradation | EbA/EbM | NA |
| SC5-10-2016—Multi-stakeholder dialogue platform to promote innovation with nature to address societal challenges | CSA | ThinkNature—Development of a multi-stakeholder dialogue platform and Think Tank to promote innovation with nature-based solutions (2016–2019): develop a NbS platform that will support the understanding and the promotion of NbS at local, regional, EU, and International levels. https://cordis.europa.eu/project/id/730338 (accessed on 28 September 2021) https://www.think-nature.eu (accessed on 1 July 2021) | 3,569,789 | GR/UK, FI, GR, IT, CH, BE, NL, FR/18 partners (8 countries) | Global | Human well-being, energy poverty, impacts of climate change | EbA/EbM, BGI/GBI, GI, BI, EE | NA |

<center>**Table A1.** *Cont.*</center>

| | Type of Call/Funding Scheme | Funded Projects/Duration/Scope/Web Portal | Budget (€) | Lead Country/Partners Countries/No. of Partners | Study Area/Scale | Challenged Addressed | NbS-Related Concept | Type of NbS Developed/Tested |
|---|---|---|---|---|---|---|---|---|
| SC5-09-2016—Operationalizing insurance value of ecosystems | RIA | NAIAD—Nature insurance value: assessment and demonstration (2016–2020): operationalize the insurance value of ecosystems to reduce the human and economic cost of risks associated with floods and drought by developing and testing the concepts, tools, applications, and business models for its mainstreaming. https://cordis.europa.eu/project/id/730497 (accessed on 2 July 2021) www.naiad2020.eu (accessed on 2 July 2021) | 5,081,176 | ES/FR, UK, ES, PO, SI, IT, DE, SE, RO, DK, NL/23 partners (10 countries) | Europe | Flood, drought | GBI/BGI, GI, EE | Green (e.g., permeable pavements, bioswales, green roofs, open retention basins, rain gardens, Façade gardens, green strips and swales). Blue (e.g., Retention ponds) Hybrid (e.g., expansion of central green Space, separated sewer for water collection and distribution) |
| SCC-04-2016: Sustainable urbanization | ERA-NET Co-funded | EN-SUGI—ERA-NET sustainable urbanization global initiative (15 projects funded) (2016–2022): bring together the fragmented R&I expertise across Europe and beyond to find innovative new solutions to the food-energy-water Nexus challenge. https://cordis.europa.eu/project/id/730254 (accessed on 28 September 2021) https://www.era-learn.eu/network-information/networks/en-sugi (accessed on 2 July 2021) https://jpi-urbaneurope.eu/news/the-15-projects-that-will-take-on-the-food-water-energy-nexus (accessed on 2 July 2021) | 18,649,260 | UK/NL, AT, DE, RO, BE, FR, SI, CY, LV, NO, TR, AR, SE, PO/20 partners (15 countries) | Europe and Argentina | Food, energy, water | GI, BI, GBI/BGI, EE, EbA/EbM | Green (e.g., Vertical greening) Blue Hybrid (e.g., food and urban agriculture) |

| | Type of Call/Funding Scheme | Funded Projects/Duration/Scope/Web Portal | Budget (€) | Lead Country/Partners Countries/No. of Partners | Study Area/Scale | Challenged Addressed | NbS-Related Concept | Type of NbS Developed/Tested |
|---|---|---|---|---|---|---|---|---|
| SCC-03-2016—New governance, business, financing models and economic impact assessment tools for sustainable cities with nature-based solutions (urban re-naturing) | RIA | Nature4Cities—Nature-based solutions for re-naturing cities: knowledge diffusion and decision support platform through new collaborative models (2016–2021): develop modules to engage urban stakeholders about re-naturing cities, develop and circulate business, financial, and governance models for NbS projects, as well as provide tools for the impact's assessment. https://cordis.europa.eu/project/id/730468 (accessed on 3 July 2021) https://www.nature4cities.eu (accessed on 3 July 2021) | 7,499,981 | FR/ES, LU, FR, HU, TR, IT, AT, UK, NL/28 partners (9 countries) | Europe | Urban challenges | GI, BI, GBI/BGI, EbA/EbM, EE | Green Blue Hybrid |
| | | NATURVATION—Nature-based urban innovation (2016–2021): develop understanding of what NbS can achieve in cities, examine how innovation can be fostered, and contribute to realizing the potential of NbS for responding to urban sustainability challenges. https://cordis.europa.eu/project/id/730243 (accessed on 2 July 2021) https://naturvation.eu (accessed on 2 July 2021) | 7,797,877 | UK/HU, DE, NL, SE, ES, UK/14 partners (6 countries) | Europe | Urban sustainability challenges | GI, BI, GBI/BGI, EbA/EbM, EE | Green Blue Hybrid |

Table A1. *Cont.*

| | Type of Call/Funding Scheme | Funded Projects/Duration/Scope/Web Portal | Budget (€) | Lead Country/Partners Countries/No. of Partners | Study Area/Scale | Challenged Addressed | NbS-Related Concept | Type of NbS Developed/Tested |
|---|---|---|---|---|---|---|---|---|
| INNOSUP-02-2016—European SME innovation Associate—pilot | CSA | INNOV—Automate VertECO (2017–2018): Create an indoor, customized green wall system designed to significantly reduce drinking water consumption by providing a plant-based technology. https://cordis.europa.eu/project/id/739732 (accessed on 2 July 2021) https://www.alchemia-nova.net/projects/automate-verteco (accessed on 2 July 2021) | 119,225 | AT/1 partner (1 country) | Household/Austria | Drinking water consumption Storm and grey water quality | GI | Hybrid |
| SC5-23-2016-2017—Support to confirmed Presidency events (conferences)—Malta, United Kingdom, Estonia | CSA | NBS2017—NbS: from innovation to common-use (2017–2018): strengthen synergy among various recent initiatives and programs launched by the EC and the Member States and develop recommendations for future practical solutions and actions. https://cordis.europa.eu/project/id/769003 (accessed on 2 July 2021) https://www.nbs2017.eu (accessed on 2 July 2021) | 274,517 | EE/EE/2 partners (1 country) | National and EU level | Natural capital Resource-efficiency and innovation Health and well-being Earth's natural limit | GI, BI, GBI/BGI, EbA/EbM, EE | NA |

Table A1. *Cont.*

| | Type of Call/Funding Scheme | Funded Projects/Duration/Scope/Web Portal | Budget (€) | Lead Country/Partners Countries/No. of Partners | Study Area/Scale | Challenged Addressed | NbS-Related Concept | Type of NbS Developed/Tested |
|---|---|---|---|---|---|---|---|---|
| MSCA-IF-2017—Individual Fellowships | SE—Society and Enterprise panel | ADAFARM—Small-scale farmers' sustainable adaptation strategies to climate change based on ecosystem services (2018–2020): analyze ecosystem-based climate adaptation options and NbS for small farmers in sub-Saharan Africa. https://cordis.europa.eu/project/id/798867 (accessed on 2 July 2021) https://www.icatalist.eu/adafarm (accessed on 2 July 2021) | 170,122 | ES/1 partner (1 country) | Small farmers in sub-Saharan Africa | Climate change | EbA | Hybrid |
| SCC-02-2016-2017—Demonstrating innovative nature-based solutions in cities | IA | CLEVER Cities—Co-designing Locally tailored Ecological solutions for Value added, socially inclusive Regeneration in Cities (2018–2023): Co-create, implement, and manage locally tailored NbS to deliver social, environmental, and economic improvements for urban regeneration, make the interventions in front-runner cities cases for successful NbS, and prepare robust replication roadmaps in fellow cities. https://cordis.europa.eu/project/id/776604 (accessed on 4 July 2021) http://clevercities.eu (accessed on 4 July 2021) | 14,864,689 | DE/UK, IT, RS, EL, ES, SE, RO, ECU, DE, AT, BE, CN/34 partners (12 countries) | Cities/Europe, East Asia and South America | Urban regeneration challenges | GI, BI, GBI/BGI, EE, | Green (e.g., green corridor, green roofs and walls) Blue (e.g., design and implement NbS around the river) Hybrid (e.g., hybrid public space and public park) |

**Table A1.** *Cont.*

| Type of Call/Funding Scheme | Funded Projects/Duration/Scope/Web Portal | Budget (€) | Lead Country/Partners Countries/No. of Partners | Study Area/Scale | Challenged Addressed | NbS-Related Concept | Type of NbS Developed/Tested |
|---|---|---|---|---|---|---|---|
| | CONNECTING Nature—Coproduction with nature for city transitioning, innovation, and governance (2017–2022): co-develop the policy and practices necessary to scale-up urban resilience, innovation, and governance via NbS. https://cordis.europa.eu/project/id/730222 (accessed on 4 July 2021) https://connectingnature.eu (accessed on 4 July 2021) | 11,699,286 | IE/BE, UK, PO, ES, IT, BG, EL, CY, BA, AM, NL, DE, IE, RO, GE, SI, CN, SK/34 partners (18 countries) | Europe, China, and South Korea | Climate change Health and well-being Social cohesion | GI, BI, BGI/GBI, EbA/EbM, EE | Green Blue Hybrid |
| | EdiCitNet—Edible cities network integrating edible city solutions for social resilient and sustainably productive cities (2018–2023): leverage the benefits that the edible city solutions effect today and catalyze their replication by launching a fully open and participatory network of cities, empowering their inhabitants by a common methodology. https://cordis.europa.eu/project/id/776665 (accessed on 4 July 2021) https://www.edicitnet.com (accessed on 4 July 2021) | 11,706,588 | DE/NL, DE, NO, ES, UK, SI, TN, TGO, URY, AT, FR, CN/35 partners (12 countries) | 12 Cities/Europe, Central America, Africa and East Asia | Urban landscapes for food production | EE | Hybrid (e.g., community gardening) |

**Table A1.** *Cont.*

| Type of Call/Funding Scheme | Funded Projects/Duration/Scope/Web Portal | Budget (€) | Lead Country/Partners Countries/No. of Partners | Study Area/Scale | Challenged Addressed | NbS-Related Concept | Type of NbS Developed/Tested |
|---|---|---|---|---|---|---|---|
| | GROW GREEN—Green cities for climate and water resilience, sustainable economic growth, healthy citizens, and environments (2017–2022): demonstrate a replicable approach for the development and implementation of city NbS strategies. https://cordis.europa.eu/project/id/730283 (accessed on 28 September 2021) http://growgreenproject.eu (accessed on 4 July 2021) | 11,519,299 | UK/ES, UK, PO, HR, IT, FR, BE, CH, NL, CN/25 partners (10 countries) | Seven cities/Europe and China | Climate change, flooding, and heat stress | GI GBI EbA/EbM | Green (e.g., Valencia Green wall) Hybrid (e.g., Manchester West Gorton Community Park, Wroclaw downtown and Great Island) |
| | URBAN GreenUP—New strategy for re-naturing cities through nature-based solutions (2017–2022): obtaining a tailored methodology to support the co-development of renaturing urban plans, and to assist in the implementation of NbS in an effective way. https://cordis.europa.eu/project/id/730426 (accessed on 14 July 2021) http://www.urbangreenup.eu (accessed on 14 July 2021) | 14,791,003 | ES/UK, ES, TR, IT, VT, PT, CO, CN/26 partners (8 countries) | Eight cities/Europe, Colombia, China, and Vietnam | Climate change Water management | EbA/EbM, BI, GI, BGI/GBI | Green Blue Hybrid |

Table A1. *Cont.*

| Type of Call/Funding Scheme | Funded Projects/Duration/Scope/Web Portal | Budget (€) | Lead Country/Partners Countries/No. of Partners | Study Area/Scale | Challenged Addressed | NbS-Related Concept | Type of NbS Developed/Tested |
|---|---|---|---|---|---|---|---|
| | proGIreg—Productive green infrastructure for post-industrial urban regeneration (2018–2023): develop NbS which are citizen-owned and co-developed by state, market, and civil society stakeholders. https://cordis.europa.eu/project/id/776528 (accessed on 14 July 2021) http://www.progireg.eu (accessed on 14 July 2021) | 11,667,247 | DE/IT, DE, HR, PT, EL, RO, BA, AT, ES, CN/34 partners (10 countries) | Seven cities/Europe | Post-industrial regeneration Lack quality greenspaces | GI, BI, GBI/BGI, EE, EbA/EbM | Hybrid (e.g., regenerating industrial soils biotic compounds, creating community-based urban agriculture and aquaponics, and making renatured river corridors) |
| | UNALAB—Urban nature labs (2017–2022): develop a robust evidence base and European framework of innovative, replicable, and locally attuned NbS to enhance the climate and water resilience of cities. https://cordis.europa.eu/project/id/730052 (accessed on 16 July 2021) https://www.unalab.eu (accessed on 16 July 2021) | 14,278,699 | FI/DE, NL, IT, FI, NO, ES, FR, CZ, TR, BE, PT, DE, SE, CN, AR/28 partners (15 countries) | 10 cities/Europe, China, and Argentina | Climate change Water management | GI, BI, GBI/BGI | Green Blue Hybrid |
| | URBiNAT—Healthy corridors as drivers of social housing neighborhoods for the co-creation of social, environmental, and marketable NbS (2018–2023): co-plan a healthy corridor as an innovative and flexible NbS, which itself integrates a large number of micro-NbS emerging from community-driven design processes. https://cordis.europa.eu/project/id/776783 (accessed on 14 July 2021) http://urbinat.eu (accessed on 14 July 2021) | 13,742,229 | PT/FR, PT, BG, IT, BE, SI, DK, SE, ES, DE, IR, CN/28 partners (12 countries) | Cities/Europe, Iran, China, Brazil, Oman, Japan | Deprived social housing Public spaces Health and well-being | EE | Hybrid (e.g., a healthy corridor as an innovative and flexible NbS, which itself integrates a large number of micro-NbS emerging from community-driven design processes) |

**Table A1.** *Cont.*

| | Type of Call/Funding Scheme | Funded Projects/Duration/Scope/Web Portal | Budget (€) | Lead Country/Partners Countries/No. of Partners | Study Area/Scale | Challenged Addressed | NbS-Related Concept | Type of NbS Developed/Tested |
|---|---|---|---|---|---|---|---|---|
| H2020-WIDESPREAD-05-2017-Twinning | CSA | ReNAture—Promoting research excellence in nature-based solutions for innovation, sustainable economic growth, and human well-being in Malta (2018–2021): establish and implement a strategy and research cluster to step-up and stimulate scientific excellence and innovation capacity in the area of NbS for sustainable development. https://cordis.europa.eu/project/id/809988 (accessed on 18 July 2021) http://renature-project.eu (accessed on 18 July 2021) | 995,905 | MT/IE, IT, UK, BG/6 partners (5 countries) | Europe | Economic growth Human well-being Environmental challenges | GI, BI, GBI/BGI, EE | NA |
| MSCA-IF-2017—Individual Fellowships | GF—Global Fellowships | Mind4Stormwater—Innovative stormwater asset management in future cities (2018–2021): help cities achieve sustainable management of their stormwater control measures. https://cordis.europa.eu/project/id/786566 (accessed on 18 July 2021) https://mind4stormwater.org (accessed on 18 July 2021) | 270,918 | FR/1 partner (1 country) | Australia and France | Stormwater management and preserving water quality | Infiltration trenches, bioretention systems | NA |

<div align="center">

**Table A1.** *Cont.*

</div>

| | Type of Call/Funding Scheme | Funded Projects/Duration/Scope/Web Portal | Budget (€) | Lead Country/Partners Countries/No. of Partners | Study Area/Scale | Challenged Addressed | NbS-Related Concept | Type of NbS Developed/Tested |
|---|---|---|---|---|---|---|---|---|
| SC5-18-2018—Valuing nature: mainstreaming natural capital in policies and in business decision-making | CSA | We Value Nature (2018–2022): establish, support, and energize an EU Valuing Nature Network of Networks and to implement a prioritized EU Valuing Nature Program, build synergies and collaborations among relevant existing and emerging networks, accelerate mainstreaming and operationalization of natural capital. https://cordis.europa.eu/project/id/821303 (accessed on 18 July 2021) https://wevaluenature.eu (accessed on 18 July 2021) | 2,192,426 | UK/CH, BE, NL/5 partners (4 countries) | Europe and global | Limited natural ecosystems | GI, BI, GBI/BGI, EE, EbA/EbM, Natural capital accounting | Green Blue Hybrid |
| SC5-08-2017—Large-scale demonstrators on nature-based solutions for hydro-meteorological risk reduction | IA | OPERANDUM—Open-air laboratories for nature-based solutions to manage environmental risks (2018–2022): reduce hydro-meteorological risks in European territories through co-designed, co-developed, deployed, tested, and demonstrated innovative green and blue/grey/hybrid NbS, and push business exploitation. https://cordis.europa.eu/project/id/776848 (accessed on 19 July 2021) https://www.operandum-project.eu (accessed on 18 July 2021) | 14,696,502 | IT/FI, NL, DE, EL, IE, UK, FR, AT, IT, ES, SK, HK, CN, AU/26 partners (13 countries) | Europe, China, Australia | Hydro-meteorological risks | GI, BI, BGI/GBI | Green Blue Hybrid |

**Table A1.** *Cont.*

| Type of Call/Funding Scheme | Funded Projects/Duration/Scope/Web Portal | Budget (€) | Lead Country/Partners Countries/No. of Partners | Study Area/Scale | Challenged Addressed | NbS-Related Concept | Type of NbS Developed/Tested |
|---|---|---|---|---|---|---|---|
| | PHUSICOS—According to nature—solutions to reduce risk in mountain landscapes (2018–2023): demonstrates how NbS provide robust, sustainable, and cost-effective measures for reducing the risk of extreme weather events in rural mountain landscapes. https://cordis.europa.eu/project/id/776681 (accessed on 18 July 2021) https://phusicos.eu (accessed on 18 July 2021) | 9,633,000 | NO/IT, DE, FR, AT, CH, NO, ES/15 partners (7 countries) | 3 large-scale demonstration sites and 2 small-scale concept cases in rural and mountainous regions/Europe | Hydro-meteorological hazards (flooding, erosion, landslides, and drought) | EbA/EbM | Hybrid |
| | RECONECT—Regenerating ecosystems with nature-based solutions for hydro-meteorological risk reduction (2018–2024): contribute to European reference framework on NbS by demonstrating, referencing, and upscaling large-scale NbS and by stimulating a new culture for 'land use planning' that links the reduction of risks with local and regional development objectives in a sustainable way. https://cordis.europa.eu/project/id/776866 (accessed on 18 July 2021) https://reconnect-europe.eu (accessed on 18 July 2021) | 15,399,379 | NL/DE, NL, UK, BE, TW, AT, DK, ES, MY, IT, PO, RS, HR, TH, BG, FR, CH, SE/37 partners (18 countries) | Local, watershed/regional level/Europe, China, Malaysia, Thailand | Hydro-meteorological risks | EbA/EbM | Hybrid |

**Table A1.** *Cont.*

| | Type of Call/Funding Scheme | Funded Projects/Duration/Scope/Web Portal | Budget (€) | Lead Country/Partners Countries/No. of Partners | Study Area/Scale | Challenged Addressed | NbS-Related Concept | Type of NbS Developed/Tested |
|---|---|---|---|---|---|---|---|---|
| SwafS-15-2018-2019—Exploring and supporting citizen science | RIA | MICS—Developing metrics and instruments to evaluate citizen science impacts on the environment and society (2019–2021): support NbS research by developing strategies and tools to evaluate impacts on science and society resulting from the integration of citizen science. https://cordis.europa.eu/project/id/824711 (accessed on 18 July 2021) https://mics.tools (accessed on 18 July 2021) | 1,944,428 | UK/NL, IT, HU, UK, RO/6 partners (5 countries) | West-East EU | Societal challenges in general | EE, BGI/GBI,EbA/EbM | Hybrid |
| SC5-13-2018—Strengthening EU–China cooperation on sustainable urbanization: nature-based solutions for restoration and rehabilitation of urban ecosystems | vspace-56ptRIA | CLEARING HOUSE—Collaborative learning in research, information-sharing, and governance on how urban tree-based solutions support Sino-European urban futures (2019–2023): investigate the role of urban forests as NbS, which refers to all measures a city can take to address urban sustainable development challenges by planting and managing trees. https://cordis.europa.eu/project/id/821242 (accessed on 19 July 2021) https://clearinghouseproject.eu (accessed on 18 July 2021) | 7,687,864 | FI/CN, PO, DE, ES, FR, IT, FI, AT, HR, BE, HK, CH/26 partners (12 countries) | Cities/Europe and China | Lack of adequate green infrastructure | GI | Green Hybrid |

Table A1. *Cont.*

| | Type of Call/Funding Scheme | Funded Projects/Duration/Scope/Web Portal | Budget (€) | Lead Country/Partners Countries/No. of Partners | Study Area/Scale | Challenged Addressed | NbS-Related Concept | Type of NbS Developed/Tested |
|---|---|---|---|---|---|---|---|---|
| | | REGREEN—Fostering nature-based solutions for smart, green, and healthy urban transitions in Europe and China (2019–2023): investigate NbS to restore, improve, enhance, and conserve natural capital and biodiversity, help build climate resilience in cities, improve liveability, and contribute to building inclusive communities. https://cordis.europa.eu/project/id/821016 (accessed on 18 July 2021) https://www.regreen-project.eu (accessed on 18 July 2021) | 5,296,191 | DK/UK, SE, DE, AT, FR, HR, DK, CN/23 parterns (8 countries) | Cities/Europe and China | Climate change Extreme weather events Biodiversity loss | EbA/EbM | Green Blue Hybrid |
| LC-CLA-09-2019—ERA-NET Co-funded action on biodiversity and climate change: Impacts, feedbacks, and nature-based solutions for climate change adaptation and mitigation | ERA-NET Co-funded | BiodivClim—Promote coordinated international research and research programs coordination to provide policymakers and other stakeholders with the tools and solutions to improve the conservation & sustainable use of biodiversity & ecosystems under a changing climate (2019–2024). https://cordis.europa.eu/project/id/869237 (accessed on 18 July 2021) https://www.biodiversa.org/1785 (accessed on 18 July 2021) | 15,151,516 | BE/AT, BE, BG, CZ, DK, EE, FI, FR, DE, EL, IE, IL, LV, LT, NO, PO, PT, RP, SK, KR, ES, SE, CH, TN, TR/35 partners (25 countries) | Europe, Israel, South Africa, Tunisia | Climate change Biodiversity loss Loss of ecosystem services | EbA/EbM, EE | Hybrid |

**Table A1.** *Cont.*

| | | | | | | | | |
|---|---|---|---|---|---|---|---|---|
| SC5-13-2019—Strengthening EU-CELAC cooperation on sustainable urbanization: nature-based solutions for restoration and rehabilitation of urban ecosystems | RIA | CONEXUS—CO-producing Nature-based solutions and restored Ecosystems: transdisciplinary neXus for Urban Sustainability (2020–2024): Co-create context-appropriate NbS for ecosystems restoration and sustainable urbanization in CELAC and EU cities, using a place-based approach. https://cordis.europa.eu/project/id/867564 (accessed on 19 July 2021) https://www.conexusnbs.com (accessed on 19 July 2021) | 6,203,619 | UK/SE, NL, DE, CO, IT, PT, CL, BR, AR, ES, RO, PE/30 partners (13 countries) | São Paulo, Bogotá, Santiago, Buenos Aires, Lisbon, Barcelona and Turin | Landscape fragmentation, urban sprawl, lacking green area | EbA/EbM | Hybrid |
| | | INTERLACE—International cooperation to restore and connect urban environments in Latin America and Europe (2020–2024): Connect cities from Europe and Latin American and equip them to restore and rehabilitate (peri)urban ecosystems. https://cordis.europa.eu/project/id/869324 (accessed on 25 June 2021) https://interlace-project.eu (accessed on 25 June 2021) | 5,476,165 | DE/ECU, ES, CR, PO, BE, NO, FR, DE, CO, NL, MX/21 partners (11 countries) | Granollers (Spain) Envigado (Colombia) Portoviejo (Ecuador) Chemnitz (Germany) Krakow (Poland) CBIMA (Costa Rica) | Urban ecosystem degradation | EbA/EbM | Hybrid (e.g., restore and rehabilitate (peri)urban ecosystems. |
| SC5-14-2019—Visionary and integrated solutions to improve well-being and health in cities | IA | VARCITIES—Visionary nature-based actions for health, wellbeing & resilience in cities (2020–2025): Create a vision for future cities with the citizen and human community at the center, implement innovative ideas and add value by creating sustainable models for improving the health and well-being of citizens https://cordis.europa.eu/project/id/869505 (accessed on 15 July 2021) https://www.varcities.eu (accessed on 15 July 2021) | 11,129,570 | EL/EL, IT, MT, SI, BE, IE, SE, NL, NO/25 partners (9 countries) | Eight cities/Europe | Urbanization and air pollution Urban exacerbation of heat islands | GBI/BGI, EE, EbA/EbM | Green Blue Hybrid |

**Table A1.** *Cont.*

| | | | | | | |
|---|---|---|---|---|---|---|
| IN-HABIT—INclusive Health And wellBeing In small and medium size ciTies (2020–2025): design integrative actions that will be shaped according to the needs of local vulnerable groups in four cities, based on culture, food, art and bonds with nature and animals combined with technological and digital means. https://cordis.europa.eu/project/id/869227 (accessed on 16 June 2021) https://www.inhabit-h2020.eu (accessed on 16 June 2021) | 11,577,919 | ES/ES, LV, IT, SK, UK, DE, BE, CO/21 partners (8 countries) | Five cities/Europe and Columbia | Health and well-being | EE | Hybrid (e.g., sustainable mobility and creative square in Cordoba) |
| EuPOLIS—Integrated NBS-based Urban Planning Methodology for Enhancing the Health and Well-being of Citizens: The EuPOLIS Approach (2020–2024): deploy natural systems to enhance public health and well-being and create resilient urban ecosystems, and regenerate and rehabilitate urban ecosystems to create inclusive and accessible urban spaces. https://cordis.europa.eu/project/id/869448 (accessed on 19 June 2021) https://eupolis-project.eu (accessed on 19 June 2021) | 11,245,408 | EL/PO, RS, DK, AT, EL, UK, HU, DE, CH, CO, IT, CY, BA, CN/28 partners (14 countries) | Four cities/Europe | Low environmental quality and low biodiversity in public spaces Water-stressed resources and undervalued use of space | EE, GBI/BGI, EbA/EbM | Green Blue Hybrid |
| GOGREEN ROUTES—GO GREEN: Resilient Optimal Urban natural, Technological and Environmental Solutions (2020–2024): position European cities as world ambassadors of urban sustainability, shifts the focus of NbS towards the co-benefits to multidimensional health-termed 360-Health. https://cordis.europa.eu/project/id/869764 (accessed on 19 July 2021) https://gogreenroutes.eu (accessed on 19 July 2021) | 11,148,168 | IE/IE, DE, UK, NO, EE, IT, FI, BE, ES, FR, CN, SE, BG, MT, NL, ES, DK, GE, AT/39 partners (19 countries) | Cities/Europe and China, Mexico, Georgia | Urban mental health and well-being | GBI/BGI, EE | Hybrid (route connecting the lakes of the city's wetland complex in Burgas, urban garden in Tallinn) |

**Table A1.** *Cont.*

| | | | | | | | | |
|---|---|---|---|---|---|---|---|---|
| MSCA-IF-2019—Individual Fellowships | MSCA-IF-EF-RI—RI—Reintegration panel | Green CURIOCITY—Green cure in overheated city spaces: An investigation of childhood heat-related health impacts and protective effects of urban natural environments (2020–2022): improve knowledge about how heat exposure during pregnancy impacts birth outcome and how long-term exposure could affect children's neurodevelopment and explore the possibilities to mitigate or prevent the negative effects of heat in the context of NbS. https://cordis.europa.eu/project/id/891538 (accessed on 19 July 2021) | 172,932 | ES/1 partner (1 country) | Europe | Overheating of cities Children's health and development | GI | Green |
| INNOSUP-01-2018-2020—Cluster facilitated projects for new industrial value chains | IA | METABUILDING—Meta-clustering for cross-sectoral and cross-border innovation ecosystem building for the European construction, additive manufacturing and NbS industrial sectors' SMEs (2020–2023): joining efforts with ICT, additive manufacturing, NbS, and the recycling industry to underpin and fuel the emergence of new cross-sectoral, cross-border industrial value chains. https://cordis.europa.eu/project/id/873964 (accessed on 18 July 2021) https://www.metabuilding.com (accessed on 18 July 2021) | 5,126,625 | FR/BE, AT, ES, PT, FR, IT, HU, UK, DE/15 partners (9 countries) | More than 140 cross-sectoral, cross-border SME-led innovation ideas/projects/Europe | Lack of collaboration between construction and other industry sectors. | NA | NA |

**Table A1.** *Cont.*

| | | | | | | | | |
|---|---|---|---|---|---|---|---|---|
| SC5-23-2019—Multi-stakeholder dialogue platform to promote nature-based solutions to societal challenges: follow-up project | CSA | NetworkNature—Advancing nature-based solutions together (2020–2023): establish a European and global platform allowing all interested stakeholders to access and contribute cutting-edge, innovative knowledge and expertise on NbS. https://cordis.europa.eu/project/id/887396 (accessed on 28 June 2021)https://networknature.eu (accessed on 26 June 2021) | 2,189,834 | DE/BE, NL, FR, CH, DE/6 partners (6 countries) | Europe and Globe | Societal challenges | GI, BI, GBI/BGI, EbA/EbM, EE | Green Blue Hybrid |
| LC-CLA-06-2019—Inter-relations between climate change, biodiversity, and ecosystem services | RIA | FutureMARES—Climate Change and future marine ecosystem services and biodiversity (2020–2024): investigate NbS for climate change adaptation and mitigation, including the restoration of habitat-forming species that can buffer coastal habitats from climate change effects and improve seawater quality. https://cordis.europa.eu/project/id/869300 (accessed on 26 June 2021) https://www.futuremares.eu (accessed on 26 June 2021) | 8,555,905 | DE/DK, EL, ES, CL, UK, PT, IT, BZ, NL, DE, IL, FR, NO, SE, FI/33 partners (15 countries) | Global | Climate change Biodiversity loss | EbA/EbM | Hybrid (e.g., restoration of costal habitat-forming species) |
| | | MaCoBioS—Marine coastal ecosystems biodiversity and services in a changing world (2020–2024): develop models on interactions between climate change, biodiversity, and functions and services of marine coastal ecosystems, establish a framework to assess vulnerabilities and evaluate the effectiveness of NbS. https://cordis.europa.eu/project/id/869710 (accessed on 28 September 2021) https://macobios.eu (accessed on 27 June 2021) | 6,980, 658 | UK/FR, PT, IE, NL, ES, IT, DE, SE, UK, JM, NO/17 partners (11 countries) | Europe and Jamaica | Climate chance, marine coastal ecosystems degradation, biodiversity loss | EbA/EbM | NA |

**Table A1.** *Cont.*

| | | | | | | |
|---|---|---|---|---|---|---|
| DRYvER—Securing biodiversity, functional integrity, and ecosystem services in drying river networks (2020–2024): collect, analyze, and model data, create a novel global meta-system approach, develop strategies to mitigate climate change effects on drying river networks, and aid their adaptation mechanisms, defining new tools and guidelines. https://cordis.europa.eu/project/id/869226 (accessed on 27 June 2021) https://www.dryver.eu (accessed on 27 June 2021) | 6,702,009 | FR/DE, AT, NL, ES, FI, HU, CZ, HR, FR, UK, SI, BR, BO, EC, CN/24 partners (15 countries) | Europe, South America, and China | Climate change, drying rivers | EbA/EbM | NA |
| PONDERFUL—Pond ecosystems for resilient future landscapes in a changing climate (2020–2024): investigate how ponds can be used as NbS for climate change, evaluate the interaction and feedback between biodiversity, ecosystem services, and climate in pondscapes, develop future scenarios for pondscapes in the EU, Latin American, and Caribbean States (CELAC), where it will conduct tests. https://cordis.europa.eu/project/id/869296 (accessed on 28 September 2021) https://ponderful.eu (accessed on 27 June 2021) | 6,993,407 | ES/DE, BE, CH, ES, UK, PT, DK, SE, FR, URY/18 partners (10 countries) | EU, Latin American and Caribbean States | Climate change | EbA/EbM | NA |

**Table A1.** *Cont.*

| | | | | | | | | |
|---|---|---|---|---|---|---|---|---|
| H2020-EU.1.1.—Excellent Science—European Research Council (ERC)-2019-STG—ERC Starting Grant | ERC-STG—Starting Grant | Niche4NbS—Beyond assuming co-benefits in NbS: Applying the niche concept for optimizing social and ecological outcomes (2020–2025): develop and test a new approach that optimizes NbS co-benefit. https://cordis.europa.eu/project/id/852633 (accessed on 28 September 2021) | 1,500,000 | IL/IL/1 partner (1 country) | Urban | Urbanization | Ecological niche concept | NA |
| LC-CLA-11-2020—Innovative nature-based solutions for carbon-neutral cities and improved air quality | IA | DivAirCity—The power of diversity and social inclusion as a mean for reducing air pollution and achieving green urban nexus in climate neutral cities (2021–2025): Focuses on the urban nexus that combines people, places, peace, economic growth, climate robustness and its impact on air quality and decarbonization, co-design solutions and trace their impact in a transparent and safe way. https://cordis.europa.eu/project/id/101003799 (accessed on 5 July 2021) https://en.iia.cnr.it/project/divaircity (accessed on 5 July 2021) | 10,794,875 | ES/IT, ES, DK, PO, RO, BE, DE, UK, EL, NL/26 partners (10 countries) | Five cities/Europe | Air pollution, GHG emission | NA | NA |

**Table A1.** *Cont.*

| | | | | | | |
|---|---|---|---|---|---|---|
| Upsurge—City-centered approach to catalyze nature-based solutions through the EU Regenerative Urban Lighthouse for pollution alleviation and regenerative development (2021–2025): Present the European Regenerative Urban Lighthouse, enable cities to unlock their regenerative potential, and provide them with knowledge and guidance in regenerative transition. https://cordis.europa.eu/project/id/101003818 (accessed on 28 September 2021) https://www.katowice.eu/Strony/UPSURGE.aspx (accessed on 5 July 2021) | 9,703,462 | SI/DE, MK, ES, BE, EL, HR, IT, UK, HU, SI, AT, NL, PO/23 partners (13 countries) | Cities/Europe | Air pollution, GHG emission | NA | NA |
| JUSTNature—Activation of nature-based solutions for a just low-carbon transition (2021–2026): activation of NbS by ensuring a just transition to low-carbon cities, based on the principle of the right to ecological space. https://cordis.europa.eu/project/id/101003757 (accessed on 5 July 2021) | 10,246,806 | IT/DE, EL, MT, HU, BE, IE, NL, IT/19 partners (8 countries) | Seven Cities/Europe | Climate change, GHG emissions | NA | NA |

## Appendix B

**Table A2.** Indicators and methods to be used to measure NbS impact.

| Impact Category/Challenges Addressed | | Indicator | Unit | Examples of Methods of Assessment | Key References |
|---|---|---|---|---|---|
| Environment | Chemical | Carbon storage and sequestration in vegetation and soil | t C/year (tons of carbon removed or stored per area per year) | Using ground-based forest growth rates, housing density data, satellite-derived land cover and tree canopy cover maps. | [135] |
| | | | t (total amount of carbon (tons) stored in vegetation) | Calculating above-ground trees and herbaceous vegetation biomass, and then transforming biomass to a carbon storage. | [136] |
| | | | Mg/ha | Allometric forest models of carbon sequestration, developed using proxy data obtained from Lidar data. | [132] |
| | | Net carbon sequestration by urban forests (including GHG emissions from maintenance activities) | $t\ C\ ha^{-1}\ yr^{-1}$ | Numerical methods calculating or estimating the interactions between vegetation and pollutants at the micro-scale allometric equations that predict vegetation growth, Forest Inventory Analysis. | [137] |
| | | Reduced energy demand for heating and cooling | $t\ CO_2$/year ($CO_2$ emissions reduced per year) | With reference to a baseline situation, the energy not consumed can be accounted for as a reduction of $CO_2$ emissions. | [138] |
| | | Annual amount of pollutants captured and removed by vegetation | t air pollutant(s) per $ha^{-1}$/year | "Tiwary method", map air purification using spatially explicit data on ecosystem types and characteristics (particularly LAI), and pollution distribution, Forest Inventory Analysis. | [139] |

Table A2. *Cont.*

| Impact Category/Challenges Addressed | Indicator | Unit | Examples of Methods of Assessment | Key References |
|---|---|---|---|---|
| Physical | Nutrient abatement, abatement of pollutants | % of mass removal | (Laboratory) experiment measuring of water quality, estimation of biomass/abatement capacity across different vegetation types. Estimation of biomass across different vegetation types. | [140] |
| | Increased evapotranspiration | ET | Estimation based of coefficients for plant types | [141] |
| | Temperature reduction in urban areas | min and max C°/day | Measurement (modeling) of day and night mean, max and min. temperatures, with respect to baseline values (°C) | [142] |
| | | | Measures of human comfort, e.g., ENVIMET PET (Personal Equivalent Temperature), or PMV (Predicted Mean Vote) | [143] |
| | Heatwave risks | persons/ha | Number of persons living in areas with x number of day sabove threshold day (>35 °C) and night temperatures (>20 °C). Temperature thresholds defining risk are slightly varying across regions, source: local health information systems. | [144,145] |
| | infiltration capacities | mm/h | Surface and extent of flooded areas, analysis of soil and vegetation characteristics | [146] |

**Table A2.** *Cont.*

| Impact Category/Challenges Addressed | | Indicator | Unit | Examples of Methods of Assessment | Key References |
|---|---|---|---|---|---|
| | Biological | Index of biodiversity | Species richness and composition per area | LIDAR, spatial analysis, and ecosystem services mapping. | [147] |
| Economy | Monetary values | Value of carbon sequestration by trees | $ t$^{-1}$ carbon | Measurements of gross and net carbon sequestration of urban trees based on calculation of the biomass of each measured tree (i-Tree Eco model), translated into avoided social costs of $CO_2$ emissions (USD t$^{-1}$ carbon). | [148] |
| | | Economic benefit of reduction of stormwater to be treated in public sewerage system | Cost of sewerage treatment by volume (€/m$^3$) | The avoided cost of runoff water in the sewerage treatment system can be used as one benefit created by the measure in a CBA. | [149–151] |
| | | Reduced energy demand for heating and cooling | €/kwh | With reference to a baseline situation, the costs of energy not consumed (=saved) is accounted for as a benefit. | [152] |
| | Non-monetary values | Job created | Number of jobs | Number of jobs created from public employment records, number of jobs in specific sectors. | [153] |
| Social | Direct social benefits | Number of users and public awareness | €, *n* of visitors/year | Contingent valuation method, travel cost, counting visitors, qualitative approaches | [154] |
| | | % of accessible public green space per capita | m$^2$/person | GIS mapping and analysis, including nearest neighbor analysis. | [155] |
| | | % of citizens living within a given distance from accessible public green space | Persons | GIS mapping using network analysis to take into account existing barriers and access ways, statistics | [156] |

**Table A2.** *Cont.*

| Impact Category/Challenges Addressed | Indicator | Unit | Examples of Methods of Assessment | Key References |
|---|---|---|---|---|
| | The availability and distribution of different types of parks and/or ecosystem services with respect to specific individual or household socioeconomic profiles and landscape design. | e.g., mean distance (or time to reach) parks per inhabitant. | Statistics GIS, definition of criteria for park types index for spatial distribution, network analysis using GIS for assessing accessibility of parks. | [157] |
| Physiological benefits | Security against violent assault, including indicators of crime bytime of day. | No. of cases/year | Statistics and perceived levels of crime and safety. | [158] |
| | Being able to participate effectively in political choices that govern one's life, including indicators on level and quality of public participation in environmental management. | Number of connection/threshold for the definition of sufficient levels of connections | Actor-Network Analysis to better understanding how different stakeholders can bias management towards certain ecosystem services. | [159] |
| | Structural aspects—family and friendship ties | Number of connection/threshold for the definition of sufficient levels of connections | Network analysis, survey, questionnaires and interviews, sampling | [160] |
| | Chronic stress and stress-related diseases as shown in cortisol levels | Cortisol slope and average cortisol levels | Measured through repeated salivary and/or hair cortisol sampling assessing effects of nature experiences through assignment of participants to particular exercises (walk in nature for a certain time) followed by psychological assessments and assessments of affective and cognitive functioning | [161] |

**Table A2.** *Cont.*

| Impact Category/Challenges Addressed | Indicator | Unit | Examples of Methods of Assessment | Key References |
|---|---|---|---|---|
| | Increase in number and percentage of people being physically active (minimum 30 min, 3 times per week) | Days with physical activity (n) | Questionnaires to ask for the number of days on which physical activity (of sufficient exertion to raise breathing rate) reached or exceeded 30 min (e.g., over the past 4 weeks) (self-reporting) | [162] |
| | Reduced percentage of obese people and children | % | Baseline needed for rate of obesity in population/eventually: reference to median city /regional/national percentage | [163] |
| | Reduction in overall mortality and increased lifespan | Number of deaths per 1000 individuals per year | Assessing effects of nature experiences through assignment of participants to exercises (e.g., walk in nature for a certain time) followed by psychological assessments and assessments of affective and cognitive functioning. | [164] |
| | Reduction in number of cardiovascular morbidity and mortality events | Number of deaths per 1000 individuals per year; morbidity scores | Composite tools for measuring health and detailed psychometric testing. | [164,165] |

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
