# Peer review of "The Role of Nature-Based Solutions for Improving Environmental Quality, Health and Well-Being"

_sustainability, doi:10.3390/su131910950_

Round 1

Reviewer 1 Report

This manuscript is a review of the concept of Nature-based Solutions. The theme is broad, and so is the review. However, it is mostly a review of concepts and terms and, therefore, quite far from the usual reviews with overviews on research content.

The abstract is too long. Beyond not fulfilling the style cannons, it totally misses the purpose of an abstract, which should be succinct, clear, and informative.

The introduction is adequate.

A review article is not a free data collection, and this is a major issue with this article.

A review article comprises a systematic review and requires a research method. For example, Identification, Screening, Sorting, Eligibility, Assessment, Information Extraction, Qualitative Synthesis, and Discussion Stages should be performed.

The temporal scope should be defined and justified for this specific case. Research keywords should be disclosed and, for each stage, it should be stated which entries were admitted and discarded.

A starting point would be consulting and adapting the PRISMA statement principles to this particular case. However, there are many recent review articles with a fully developed methodology. I recommend a deeper investigation on this matter.

The conclusions section should be much more concise and clearer.

Redaction is not bad, but it lacks proficiency. There are some typos and sentences such as “NbS is steady rising with recent and intense rise since 2016” that should be re-written. I recommend a full revision.

Referencing does not seem to comply with the Journal standards.

Figure 1 text is hardly readable. Please reformulate.

Author Response

Please see attached response to reviewer.

Reviewer 2 Report

The study summarized the definitions, development and applications of NbS. The concepts and definitions of NbS and the related terminologies were discussed in a relatively comprehensive way, but the implications and contributions were insufficient and still need to be highlighted. The specific comments are as follows:

1. Page 2 Line 79-99: the authors are suggested to added a figure for the reminders to make the structure of this paper directly and clearly, but it is not a compulsive requirement.

2. The abbreviations of the terminology, such as “IUCN”, should give their full names when they first appear.

3. Page 4 Line 141-145: is there any reference or evidence to support this statement?

4. Page 4 Line 183-185: is there any reference or evidence to support this statement?

5. Page 5 Line 197-199: how to ensure that all the papers are related to the themes, is there any method to exclude irrelevant research?

6. Page 6 section 2.3 and 2.4: the authors conducted some descriptive statistics about the trends, what is the implication for these descriptive statistics?

7. Page 8 section 3.1: it is not so easy to identify the framework of nternational Intra-governmental and Non-govermental Governance, please describe it in a clearer way. And more discussions in-depth such as the heterogeneity or consistency of the two part are expected.

8. Page 9 Line 278-310: please show the method or criteria used for classifying these projects, and it seems that there are some overlaps in the current classification, like type ii and iii, type i and vii.

9. Page 11 section 3.5: it seems that the NbS has wide applications in environment, society and economy. But in fact, it is quite difficult to solve these issues simultaneously, so the limitations of NbS itself need to be discussed in a proper position.

10. The novelty and contributions of this paper are relatively insufficient; the authors should better describe the contributions of this paper and highlight its novelty.

Author Response

(The authors gave the same response as above.)

Round 2

Reviewer 1 Report

I thank the Authors for the rebuttal and some enhancements.

I find surprising the will to position this article as a research one, considering the innovative content that a research article must bring, but I believe that all my comments have been addressed.

As a research article, this manuscript should identify a shortcoming in the current body of knowledge and find a novel solution, employing a well depicted and replicable methodology. 

Author Response

Dear reviewer,

Thank you very much for your thoughtful comments and useful suggestions. 

Please see attached file for our responses to your review. 

Kind regards

Hai-Ying Liu

Reviewer 2 Report

The authors have addressed all my questions.

Author Response

Hi, 

Thank you so much for spending time to review our manuscript, and for your very valuable comments. 

Kind regards 

Hai-Ying on behalf of all co-authors